# Binding Mechanism of CD47 with SIRPα Variants and Its Antibody: Elucidated by Molecular Dynamics Simulations

**DOI:** 10.3390/molecules28124610

**Published:** 2023-06-07

**Authors:** Kaisheng Huang, Yi Liu, Shuixiu Wen, Yuxin Zhao, Hanjing Ding, Hui Liu, De-Xin Kong

**Affiliations:** 1State Key Laboratory of Agricultural Microbiology, Agricultural Bioinformatics Key Laboratory of Hubei Province, College of Informatics, Huazhong Agricultural University, Wuhan 430070, China; kshuang@webmail.hzau.edu.cn (K.H.);; 2School of Basic Medical Sciences, Hubei University of Science and Technology, Xianning 437100, China

**Keywords:** tumor immune response, CD47-SIRPα, B6H12.2, molecular dynamics simulation, binding mechanism, hotspot residue, inhibitor design

## Abstract

The intricate complex system of the differentiation 47 (CD47) and the signal-regulatory protein alpha (SIRPα) cluster is a crucial target for cancer immunotherapy. Although the conformational state of the CD47-SIRPα complex has been revealed through crystallographic studies, further characterization is needed to fully understand the binding mechanism and to identify the hot spot residues involved. In this study, molecular dynamics (MD) simulations were carried out for the complexes of CD47 with two SIRPα variants (SIRPαv1, SIRPαv2) and the commercially available anti-CD47 monoclonal antibody (B6H12.2). The calculated binding free energy of CD47-B6H12.2 is lower than that of CD47-SIRPαv1 and CD47-SIRPαv2 in all the three simulations, indicating that CD47-B6H12.2 has a higher binding affinity than the other two complexes. Moreover, the dynamical cross-correlation matrix reveals that the CD47 protein shows more correlated motions when it binds to B6H12.2. Significant effects were observed in the energy and structural analyses of the residues (Glu35, Tyr37, Leu101, Thr102, Arg103) in the C strand and FG region of CD47 when it binds to the SIRPα variants. The critical residues (Leu30, Val33, Gln52, Lys53, Thr67, Arg69, Arg95, and Lys96) were identified in SIRPαv1 and SIRPαv2, which surround the distinctive groove regions formed by the B2C, C’D, DE, and FG loops. Moreover, the crucial groove structures of the SIRPα variants shape into obvious druggable sites. The C’D loops on the binding interfaces undergo notable dynamical changes throughout the simulation. For B6H12.2, the residues Tyr32_LC_, His92_LC_, Arg96_LC_, Tyr32_HC_, Thr52_HC_, Ser53_HC_, Ala101_HC_, and Gly102_HC_ in its initial half of the light and heavy chains exhibit obvious energetic and structural impacts upon binding with CD47. The elucidation of the binding mechanism of SIRPαv1, SIRPαv2, and B6H12.2 with CD47 could provide novel perspectives for the development of inhibitors targeting CD47-SIRPα.

## 1. Introduction

Over the past decade, immunotherapy has emerged as a significant cancer treatment [1,2,3]. One of the critical factors in tumor immune escape is the inhibition of specific immune checkpoints [2]. By blocking these immune checkpoints, an anticancer immune response can be activated, bringing promising prospects to cancer treatment [1,3].

CD47 (the cluster of differentiation 47) is a self-protective immunoglobulin and is widely expressed in both normal and cancerous cells [4,5,6]. As the primary receptor of CD47, SIRPα (the signal-regulatory protein alpha) is mainly found on the surface of myeloid cells such as macrophages [7]. The binding of CD47 and SIRPα leads to the inhibition of phagocytic activity in macrophages, which is beneficial for normal cells but also allows cancer cells to evade immune detection and elimination [8,9,10]. This phenomenon has been observed in many tumor cells, e.g., the increased expression of CD47 is associated with poor prognosis in leukemia [11,12]. Therefore, the CD47-SIRPα complex is considered to be a critical target for immune checkpoint blocking [13].

Studies on the clinical characteristics of the CD47-SIRPα axis have been partially reported [14]. Furthermore, the crystal structures of CD47 with SIRPαv1 and SIRPαv2 have been elucidated [15,16], as well as the structures of antibodies targeting CD47-SIRPα [17,18]. The N-terminal domain of SIRPα exhibits high polymorphism in comparison to its main ligand CD47, which shows minimal variation [19,20,21]. Current understandings of the CD47-SIRPα axis in biological mechanisms indicate that CD47-SIRPα has broad prospects in the field of immunotherapy [22,23]. Experiments in vitro have revealed that CD47 has a critical role for almost all solid tumor cells of humans to inhibit phagocytic activity and avoid elimination [24]. The anti-SIRPα therapy also has encouraging outcomes (less blood toxicity) in preclinical investigations, while its clinical implementation is hindered by the existence of various SIRPα variants [25].

Prior investigations into CD47-SIRPα primarily focused on crystallographic structure analysis and biochemistry experiments that aimed at uncovering its clinical mechanisms. However, these approaches do not offer comprehensive insights into the dynamic interaction processes of the CD47-SIRPα system. Furthermore, the development of CD47-SIRPα drugs, especially the development of small molecule suppressants, is still insufficient. The understanding and revealing of the dynamical binding mechanism can provide valuable information for drug discovery of CD47-SIRPα. The intricate details of interactions that may be overlooked in clinical experiments can be revealed through molecular dynamics (MD) simulations, offering valuable understandings for the binding mechanisms and assisting in the designs of novel inhibitors [26,27]. This has been demonstrated in previous studies and is a highly sought-after outcome by researchers in the field [28,29,30].

Although X-ray structure analysis provided static crystallographic information on the binding surface of CD47-SIRPαv1, CD47-SIRPαv2, and CD47-B6H12.2 [15,16,17], the state of the binding interfaces in equilibrium and the energy contributions of the residues at the binding interface are still unclear. By analyzing the structure of complexes in equilibrated state, the druggable sites in MD simulation can be characterized. Crystallographic studies have also provided structural information such as static hydrogen bonds [15], while the dynamical features of structures at the binding interfaces need to be further characterized. MD simulations can explore the dynamical changes of proteins during the interaction process, which is a unique advantage for MD simulations in studying the binding mechanism compared with methods such as crystallographic experimental research [31,32].

In this study, the structural and thermodynamic features of CD47 in complex with the two most common SIRPα variants (SIRPαv1, SIRPαv2) [15,16] and with the anti-CD47 antibody B6H12.2 [17] were investigated using MD simulations. The binding mechanisms were systematically elucidated by multiple methods, including free energy calculation, dynamical cross-correlation matrix analysis, druggable site characterization, dynamical residue contact analysis, normal mode analysis, hydrogen bond calculation, definition of secondary structure analysis, etc. The dynamical changes of proteins were revealed. The energetic and structural features of the three complexes were analyzed, and the critical regions and hotspot residues in the complexes were identified. We expected that the study could demonstrate the probable binding mechanisms of CD47 with SIRPαv1, SIRPαv2, and B6H12.2 and provide potential inspirations for designing inhibitors.

## 2. Results

### 2.1. Static Structural Information Analysis

The crystallographic structures and the sequence information of CD47-SIRPαv1, CD47-SIRPαv2, and CD47-B6H12.2 are shown in Figure 1 and Figure 2. According to our preliminary judgment of the crystallographic structures and the sequence information, the CD47 proteins in three complexes are consistent (mean RMSD = 0.365 Å).

There are 13 amino acid mutations between SIRPαv1 and SIRPαv2 (RMSD = 0.386 Å). Crystallographic studies suggested that most of these mutations are located far away from the binding interface of CD47-SIRPαv1 and CD47-SIRPαv2, and that the mutations do not seem to affect the binding significantly [16]. The corresponding secondary structures in the proteins were marked with letters in Figure 1 and Figure 2.

A distance threshold of 5.0 Å was used to analyze the static binding interfaces in CD47-SIRPαv1, CD47-SIRPαv2, and CD47-B6H12.2. It could be found that there are multiple residues on the binding interfaces of the three complexes. The residue information at the interfaces of the crystal structures was summarized in Table 1. These residues clearly exhibit the initial residue distributions of the protein interfaces. However, the residues in static states provide only a shallower understanding of the protein binding interfaces. The dynamically energetic and structural features of these residues during the protein binding process require more intensive studies. MD simulations can further explain the binding mechanism of proteins and reveal the energetic features of the residues on the protein binding interface. Thus, MD simulations were executed to further characterize the efforts of the residues in the static interfaces of the three complexes.

### 2.2. Structural Stability during Simulations

A total of 500 ns simulations were performed on three systems, and the simulation of each system was conducted in triplicate. RMSD and RMSF were calculated to explore the structural fluctuations and stabilities.

The RMSD results for three complexes were relatively consistent throughout the simulations. As depicted in Figure 3, the structural variations of three protein complexes are considerably slight. It can be observed that during most of the simulation time, all the three systems sustained a state of convergence and equilibrium.

RMSF were calculated using all the Cα atoms. The results in Figure 4 show that the fluctuations of the CD47 proteins in the three systems are quite similar, with the majority of the segments in CD47 displaying almost identical fluctuations, except for the AB regions. Compared with CD47 bound to the SIRPα variants, the AB region of CD47 bound to the antibody has a higher flexibility. The structural fluctuations of SIRPαv1 and SIRPαv2 are similar. Most regions of the two proteins are relatively stable, and the main flexible parts concentrate in the CC’D regions. The fluctuations of B6H12.2 mainly occurred in the second half of the heavy and light chains. The light chain showed relatively obvious flexibility around the residue Gln160_LC_, and the heavy chain showed obvious flexibility around the residue Thr140_HC_.

### 2.3. Characterization of Protein Druggable Sites

The RMSD analysis showed that all the three systems sustained a state of convergence and equilibrium during most of the simulation time. Through analyzing proteins in the equilibrated state by MOE [33], the molecular druggable sites in SIRPαv1 and SIRPαv2 were found.

According to Figure 5, it could be found that the main residues surrounding the druggable sites of SIRPαv1 and SIRPαv2 include Leu30, Val33, Gly34, Pro35, Gln52, Lys53, Arg69, and Lys96. The energetic and structural features of the druggable sites and surrounding residues of SIRPαv1 and SIRPαv2 are similar, which indicates that there may be no significant difference in the drug properties of the two proteins.

It is worth noting that these residues distribute mainly in the grooves formed by the B2C, C’D, DE, and FG loops in the SIRPα variants. The grooves formed by these four loops are the crucial feature for their interactions with CD47, providing the main binding interfaces. The druggable sites and corresponding residues distributed on the groove regions presumably possess potential research value and deserve special attention. MD simulations are able to further elucidate the binding mechanism of the residues around the druggable sites and reveal their energetic properties. By performing MD simulations on CD47-SIRPαv1 and CD47-SIRPαv2, the dynamically energetic and structural features of these residues can be systematically analyzed. Furthermore, this information can be used to gain deeper understandings of their influence during protein interactions. By combining the distribution of the druggable sites, these analyses can provide valuable insights for drug exploration.

### 2.4. Dynamic Mechanism and Structural Change

The RMSF analysis revealed that the C’D loops of SIRPαv1 and SIRPαv2 possessed more structural fluctuations than other regions throughout the simulation. The C’D loops of SIRPαv1 and SIRPαv2 are important structural regions on their binding interfaces with CD47, and the dynamical changes of the C’D loops should deserve special attention. An average of 500 frames were equally extracted from the simulated trajectories to observe the dynamical changes. The variations of the C’D loops in the SIRPα variants were analyzed by PYMOL [34].

In addition, the DSSP (Definition of Secondary Structure of Protein) analysis was performed to further observe the secondary structure changes of SIRPαv1 and SIRPαv2. The dynamic structure information is shown in Figure 6. The C’D loops in SIRPαv1 and SIRPαv2 were observed to have special conformational flips during the simulation. The structural changes of the C’D loops are worthy of attention, and they may have potential research significance for the exploration of protein binding mechanisms.

The RMSF analysis revealed that the CC’ and C’D loops possess more variations than most other regions of the SIRPα variants. According to Figure 7, the secondary structure of the CC’ loops (residues 42–46) and C’D loops (residues 52–57) in SIRPαv1 and SIRPαv2 exhibited relatively abundant fluctuations during simulations, which is consistent with the RMSF analysis of the SIRPα variants. Considering that the C’D loops of SIRPαv1 and SIRPαv2 are important structural regions during their interactions with CD47, the dynamical change of the C’D loops is particularly noteworthy, and it may have special impacts on the binding mechanism of the protein complexes. The MD simulations are able to capture the special dynamical changes of the C’D loops in protein interactions, which are difficult to be observed precisely in traditional experiments or crystal structure studies. The dynamical changes of the C’D loops in the SIRPα variants are meaningful for further elucidating the binding mechanism of CD47-SIRPαv1 and CD47-SIRPαv2. The corresponding dynamical changes and DSSP analyses of the second and third simulations are shown in Appendix A.

### 2.5. Binging Free Energy Analysis

The MM-GBSA method was used to calculate the binding free energies for the three complexes. The results are shown in Table 2.

According to the calculations, the enthalpy changes are the main driving forces behind the protein interactions in three complexes. The major contributions of the enthalpy changes are from electrostatic interactions and van der Waals forces, while nonpolar solvent contributions provide minor driving forces for binding. Calculations showed that the major energy consumption during binding comes from polar interactions between the proteins and the solvent.

Generally speaking, the protein bindings in CD47-SIRPαv1, CD47-SIRPαv2, and CD47-B6H12.2 mean that the entropy of the protein systems should become smaller. Calculations showed that the entropy changes offset part of the energy contributions, which led to decreases in the absolute values of the total binding free energy. The calculated values of entropy changes are stable, and the differences in entropy changes among the three complexes are relatively insignificant.

The calculated total Gibbs free energy of CD47 with B6H12.2 is significantly lower than that of CD47 with the SIRPα variants, which is consistent with the experiments showing a stronger affinity of CD47-B6H12.2 [16,17], suggesting that B6H12.2 should possess the ability to inhibit the binding of CD47 with SIRPα variants.

The energy contribution provided by the van der Waals interaction in CD47-B6H12.2 is consistently higher than that of CD47-SIRPαv1 and CD47-SIRPαv2 in all three simulations. Compared to CD47-SIRPαv1 and CD47-SIRPαv2, the main reason for the lower binding free energy of CD47-B6H12.2 is its relatively significant enthalpy contribution. The significant enthalpy contribution in CD47-B6H12.2 is the result of the combined effect of multiple energetic components including solute and solvent interactions. Therefore, the drug design inspiration obtained from CD47-B6H12.2 should be learned from the overall interaction mode of CD47 with B6H12.2 from the perspective of multiple energy changes. Similar results in the free energy calculations were shown in the other two replicates of the three complexes. The energy results of the second and third simulations are exhibited in Appendix A.

### 2.6. Energy Decomposition and Analysis of the Binding Hot Spots

The free energy decomposition was used to analyze the energy contribution of different residues and provide valuable information for identifying hotspots in three complexes. The identification of hotspot residues is based on a threshold of −2 kcal/mol for the energy contribution value. To be considered as a hotspot residue, the energy decomposition results for a given residue have to fall below this threshold in at least two simulations. Figure 8 and Figure 9 exhibit the residue free energy decomposition curves and the distributions of hotspot residues in proteins.

For CD47-SIRPαv1, the hotspot residues of CD47 distribute mainly in the C strand and FG region. The hotspot residues in the C strand of CD47 are Glu35 and Tyr37. The FG region possesses more hotspot residues, including Glu100, Leu101, Thr102, Arg103, and Glu104. In CD47 of CD47-SIRPαv1, the main energy contribution of Glu35 is the prominent electrostatic interaction. The electrostatic interaction of Glu104 is also significant, but the polar interaction with the solvent during the binding process brings more energy consumption, and the energy contribution of Glu104 is mainly from nonpolar interactions. The energy contribution of Thr37 is composed of both electrostatic and van der Waals interactions. The main energy contribution of Leu101, Thr102, and Arg103 comes from the van der Waals interaction, and the nonpolar hydrophobic interaction also provides part of the energy contribution. Among these hotspots, the residues Glu35, Leu101, and Thr102 possess relatively more prominent energy contributions.

The main energy contributions of SIRPαv1 concentrate in the B2C, C’D, DE, and FG loops. The hotspot residues in the B2C loop of SIRPαv1 include Leu30, Val33, Gly34, and Pro35. The hotspot residues in the C’D loop include Gln52 and Lys53. The hotspot residues in the DE loop of SIRPαv1 include Thr67 and Arg69, and the hotspot residues in the FG loop include Arg95 and Lys96. In SIRPαv1, the electrostatic interactions of residues Gln52, Lys53, Arg69, Arg95, and Lys96 are significant, but only Arg95 of these residues contributes main energy from electrostatic interactions. The energy contributions of Lys53, Thr67, Arg69, Lys96 are composed of electrostatic and van der Waals interactions. The energy contributions of Leu30, Val33, Gly34, Pro35, and Gln52 are mainly from nonpolar components. Among the hotspot residues in SIRPαv1, the energy contributions of Val33, Lys53, Thr67, Arg69, and Lys96 are relatively more significant.

The residue energy decomposition results of the complex CD47-SIRPαv2 are similar to those of CD47-SIRPαv1. In CD47-SIRPαv2, the hotspot residues in the C strand of CD47 are Glu35 and Tyr37. The hotspot residues in the FG region are Glu100, Leu101, Thr102, and Arg103. The contribution of Glu104 in CD47 of CD47-SIRPαv2 is not as significant as it was in the complex of CD47-SIRPαv1. The energy decomposition results are found to be in good agreement with the mutation experiments conducted on CD47 [35]. The residues Tyr37 and Glu100 were observed to have substantial contributions to the binding energy, which were also reflected in the significant impacts that were observed in the mutation experiments [35]. This suggests that the C strand and the FG region of CD47 could potentially play critical roles in the binding of CD47 to SIRPαv2.

In the B2C loop of SIRPαv2, the residues Leu30, Val33, and Pro35 are identified as the hotspot residues; the C’D loop contains Gln52 and Lys53 as the hotspot residues; the DE loop is characterized by Thr67 and Arg69 as the hotspot residues; and the FG loop contains Arg95 and Lys96 as the hotspot residues. The energy contribution of Gly34 in SIRPαv2 is not as significant as in SIRPαv1, while the contributions of other hotspot residues are pretty similar in the two variants. The mutant amino acids between SIRPαv2 and SIRPαv1 do not show significant influence on the energy decomposition. The outcomes of the mutation experiments and energy decomposition analyses exhibit a favorable level of consistency in crucial areas of SIRPαv2 [35]. Specifically, Val33 in the B2C loop, Arg69 in the DE loop, and Lys96 in the FG loop are found to have significant impacts both in the energy decomposition analysis and mutation experiments. The energy components of hotspots in the two proteins of CD47-SIRPαv2 are basically the same as those in CD47-SIRPαv1. In CD47 of CD47-SIRPαv2, the residues Glu35, Leu101, and Thr102 possess relatively more obvious energy contributions among the hotspot residues. Among the hotspot residues of SIRPαv2, residues with relatively more prominent energy contributions include Val33, Lys53, Arg69, and Lys96.

For CD47 in CD47-B6H12.2, the hotspot residues distribute mainly in the CC’ strand and FG region. For B6H12.2, the hotspot residues locate mainly in the first half of the heavy and light chains. The heavy chain of B6H12.2 has relatively more hotspot residues. It is worth noticing that the main distributions of hotspot residues in CD47 have no significant change in the three complexes, indicating that CD47 may tend to bind with B6H12.2, SIRPαv1 and SIRPαv2 in a similar manner. Furthermore, the energy contributions of the FG regions in CD47 are relatively more significant in all three complexes. The hotspot residues in the CC’ strand of CD47 include Thr34, Glu35, Tyr37, and Asp51. The hotspot residues in the FG region include Thr99, Glu100, Leu101, Thr102, and Glu104. In CD47 of CD47-B6H12.2, the electrostatic interactions of the hotspot residues Glu35, Asp51, Glu100, and Glu104 are significant, but only the residue Asp51 among them contributes main energy from the electrostatic interaction; The energy contributions of Glu35, Glu100, and Glu104 are composed of electrostatic and van der Waals interactions; The energy contributions of residues Thr34, Tyr37, Thr99, Leu101, and Thr102 are mainly from nonpolar components; among these hotspot residues, the energy contributions of Glu35, Leu101, and Glu104 are relatively more prominent.

The hotspot residues in the light chain of B6H12.2 include Tyr32_LC_, His92_LC_, and Arg96_LC_. The hotspot residues in the heavy chain of the antibody include Tyr32_HC_, Thr52_HC_, Ser53_HC_, Thr56_HC_, Thr59_HC_, Leu100_HC_, Ala101_HC_, and Gly102_HC_. For antibody B6H12.2, the hotspot residues His92_LC_, Arg96_LC_, Thr52_HC_, and Ser53_HC_ possess significant electrostatic interactions, among which the electrostatic interactions provide the main energy contributions in residues Arg96_LC_ and Ser53_HC_. The energy contributions of Tyr32_LC_, Thr52_HC_, Thr56_HC_, and Gly102_HC_ are composed of electrostatic and van der Waals interactions. The energy contributions of His92_LC_, Tyr32_HC_, Tyr59_HC_, Leu100_HC_, and Ala101_HC_ are mainly from nonpolar components. Among the hotspots in B6H12.2, the residues Tyr32_LC_, His92_LC_, Arg96_LC_, Thr52_HC_, Ser53_HC_, and Ala101_HC_ have relatively more prominent energy contributions.

### 2.7. Dynamical Structure Analysis

#### 2.7.1. Hydrogen Bonds in the Complexes

Table 3 shows the important hydrogen bonds in the three complexes. These hydrogen bonds are considered to be important interactions and contribute crucially to the protein association and stabilization. The hydrogen bonds that contribute significantly to the binding of the complexes are shown in Figure 10 with specific structures and residue distributions. More detailed and comprehensive information is provided in Appendix A.

According to the hydrogen bond information shown in Table 3 and Appendix A, it can be noticed that for the CD47 proteins in all three systems, the residues forming hydrogen bonds distribute mainly in the CC’ strand and FG region. The B2C, C’D, DE, and FG loops of the SIRPαv1 protein are observed to possess notable hydrogen bond interactions, and the stable hydrogen bonds exist mainly in the B2C and DE loops. The SIRPαv2 protein has hydrogen bond interactions in the B2C loops, DE loops, and F strand. Similar to SIRPαv1, the main hydrogen bonds of SIRPαv2 exist in the B2C and DE loops. For B6H12.2, the residues that form hydrogen bonds with CD47 are mainly within the first half of the heavy and light chains. The hydrogen bond interaction in the two proteins of CD47-B6H12.2 is more significant than CD47-SIRPαv1 and CD47-SIRPαv2, and the significant hydrogen bonds in CD47-B6H12.2 provide important contributions to the polar interactions between the proteins.

The vital hydrogen bonds across different complexes were scrutinized and singled out. In CD47-SIRPαv1, the residues forming the important hydrogen bonds are Arg103-Thr67 (CD47 residue–SIRPαv1 residue), Leu101-Gly34, and Glu35-Arg69. Among these hydrogen bonds in CD47 (bound to SIRPαv1), the residues Glu35, Leu101, and Arg103 also show obvious energy contributions during simulations. The side chain of Glu35 in CD47 has a significant polar energy contribution during the protein interaction, which is consistent with Glu35 participating in the formation of stable hydrogen bonds.

In the CD47-SIRPαv2 complex, the residues forming the important hydrogen bonds are Arg103-Thr67 (CD47 residue–SIRPαv2 residue), Leu101-Gly34, Glu35-Arg69, and Glu100-Arg69. Among these hydrogen bonds in CD47 (bound to SIRPαv2), the residues Glu35, Glu100, Leu101, and Arg103 exhibit obvious energy contributions during simulations. The side chains of Glu35 and Glu100 in CD47 have significant polar energy contributions during the protein interaction, which is consistent with Glu35 and Glu100 participating in the formation of important hydrogen bonds. The residues Thr67 and Arg69 are also the hotspot residues in SIRPαv1 and SIRPαv2. The side chains of Arg69 in the SIRPα variants possess significant energy contributions in the protein interaction, and the main chains of Thr67 also provide partial energy contributions, showing a good agreement between energy decomposition and hydrogen bond analysis.

In the CD47-B6H12.2 complex, the residues with the stable hydrogen bonds are Thr99-His92_LC_ (CD47 residue–B6H12.2 residue), Glu100-Gly102_HC_, Thr34-Gly31_HC_, Asp51-Gln53_LC_, Glu104-Arg96_LC_, Leu101-Arg96_LC_, Glu35-Ser53_HC_, Glu97-Tyr32_LC_, and Glu104-Arg96_LC_. Among these, the residues Glu35, Asp51, Thr99, Glu100, Leu101, Glu104 of CD47 (bound to B6H12.2) and Tyr32_LC_, His92_LC_, Arg96_LC_, and Ser53_HC_ of B6H12.2 exhibit obvious energy contributions during simulations. The side chains of Glu35, Asp51, Glu104 in CD47 and Arg96_LC_, and Ser53_HC_ in B6h12.2 possess significant polar energy contributions during the protein interactions, which is consistent with these residues participating in the formation of important hydrogen bonds.

#### 2.7.2. Salt Bridges

Table 4 shows the important salt bridges in CD47-SIRPαv1, CD47-SIRPαv2, and CD47-B6H12.2. Figure 11 exhibits the structures and spatial distributions of the salt bridges in Table 4.

The analysis of salt bridges shows that in CD47-SIRPαv1 and CD47-SIRPαv2, the residues participating in salt bridges are situated mainly in the CC’ strands and FG regions of CD47. The major salt bridges are located in the C’D, DE, and FG loops of the SIRPα variants. In CD47-B6H12.2 complexes, the salt bridges are established between the CC’ strand of CD47 and the light chain of B6H12.2.

The residue pairs that correspond to the stable salt bridges were identified. In the CD47-SIRPαv1 complex, the residues forming the stable salt bridges are Glu35-Arg69 (CD47 residue–SIRPαv1 residue), Glu97-Lys96, Glu97-Lys53, Asp51-Arg95, Glu104-Lys96, and Glu104-Lys53. For the CD47-SIRPαv2 complex, the residues forming the stable salt bridges are Glu35-Arg69 (CD47 residue–SIRPαv1 residue), Asp51-Arg95, Glu97-Lys96, Glu104-Lys53, Glu106-Lys53, and Glu97-Lys53. In CD47-SIRPαv1 and CD47-SIRPαv2, the residues Glu35, Glu104 in CD47 and Lys53, Arg69, Arg95, and Lys96 in the SIRPα variants also exhibited obvious energy contributions during simulations. Furthermore, the side chains of Glu35 in CD47 and Lys53, Arg69, Arg95, and Lys96 in SIRPαv1 and SIRPαv2 possess significant electrostatic interactions, and the prominent electrostatic interactions provide important energy contributions, which are consistent with their participation in the formation of important salt bridges.

For the CD47-B6H12.2 complex, the stable salt bridges are formed between Asp51-Lys49_LC_ and Lys39-Asp31_LC_ (CD47 residue–B6H12.2 residue). The residue Arg69 of CD47 is found to participate significantly in the salt bridge interaction of CD47-B6H12.2, which is consistent with its prominent contribution to the electrostatic interaction during the protein binding. The residue Asp51 in CD47 (bound to B6H12.2) also showed an obvious energy contribution during simulations. The side chains of Asp51 in CD47 and Lys49_LC_ in B6H12.2 possess significant polar interactions, which is consistent with these residues participating in the formation of stable salt bridges.

#### 2.7.3. Dynamical Residue Contacts

The native contacts program in CPPTRAJ was used to conduct the dynamical residue contact analysis. This analysis could provide detailed information regarding the binding mechanism of proteins.

Figure 12 shows the residues forming remarkable contacts in different complexes during protein interactions, and more detailed information can be found in Appendix A. Based on the information in Figure 12, the residues that contributed significant contacts were identified and their structures were displayed in Figure 13. Hydrogen bonds, salt bridges, and dynamical residue contacts are major binding mechanisms in the three complexes. The structural analyses can further verify the results of residue energy decomposition. The residues that play an important role in both energy analysis and structural analysis will be considered as crucial residues in protein interactions.

According to the dynamical residue contact analysis, for CD47 in the complexes of CD47 with the SIRPα variants, the contact residues situate mainly in the BC and FG regions. The contact residues of the two SIRPα variants are similar, which distribute mainly in the B2C, C’D, DE, and FG loops. The FG regions of CD47 insert into the grooves formed by Leu30, Val33, Gly34, Pro35, Gln52, Lys53, Thr67, Arg69, Arg95, and Lys96 of the SIRPα variants, contributing the main interaction interfaces in the two complexes. The BC regions of CD47 bind to the outer edges of the grooves on the sides of the B2C, DE, and FG loops and form contacts mainly with the residues in the DE and FG loops of the SIRPα variants.

In the CD47-B6H12.2 complex, the residues forming contacts in B6H12.2 concentrate mainly in the first half of the heavy and light chains. The contact residues in CD47 of CD47-B6H12.2 still concentrate mainly in the BC and FG regions. It indicates that B6H12.2 may produce steric hindrance with SIRPαv1 and SIRPαv2 due to the overlap of contact regions when they bind to CD47, which could lead to competitive inhibition. The residues Tyr32_LC_, His92_LC_, Arg96_LC_, Thr52_HC_, Tyr59_HC_, Leu100_HC_, Ala101_HC_, and Gly102_HC_ in B6H12.2 surround a structural region, and the FG region of CD47 is mainly in contact with the annular region in B6H12.2. The BC region of CD47 forms contacts mainly with the outer edge of the surrounded region on the side of Thr52_HC_, Ala101_HC_, and Tyr32_LC_.

Critical residues with significant contacts in different complexes were further identified. In CD47-SIRPαv1, the important contacting residue pairs include Tyr37-Lys96 (CD47 residue–SIRPαv1 residue), Glu29-Arg69, Leu101-Val33, Arg103-Arg69, Leu101-Leu30, Glu104-Gln52, Glu106-Lys53, Gln1-Gln52, and Glu104-Lys53. The residue pairs that formed significant contacts between CD47-SIRPαv2 and CD47-SIRPαv1 are very similar. In the CD47-SIRPαv2 complex, the closely contacted residues include Tyr37-Lys96 (CD47 residue–SIRPαv2 residue), Leu101-Val33, Glu29-Arg69, Leu101-Leu30, Arg103-Arg69, Gln1-Gln52, and Glu104-Lys53. In both CD47-SIRPαv1 and CD47-SIRPαv2, the residues Tyr37, Leu101, Arg103, Glu104 in CD47 and Leu30, Val33, Gln52 in the SIRPα variants possess obvious energy contributions during simulations, which are consistent with their stable residue contacts during the protein interaction.

The residue pairs forming intimate contacts in CD47-B6H12.2 include Tyr37-Phe50_LC_ (CD47 residue–B6H12.2 residue), Glu100-His92_LC_, Thr102-Arg96_LC_, Thr34-Tyr32_HC_, Thr102-Tyr59_HC_, Glu35-Ser53_HC_, Glu97-Tyr32_LC_, and Glu29-Tyr57_HC_. The residues Glu35, Glu104 in CD47 and Tyr32_LC_, His92_LC_, Tyr32_HC_, Tyr59_HC_ in B6H12.2 possess significant energy contributions during the protein binding, which agree well with their stable residue contacts in the protein binding process.

### 2.8. Dynamical Correlation Analysis of CD47 Proteins

The DCCM map shows the dynamical correlation of the internal motions in CD47 of the three complexes. In the DCCM maps of Figure 14, +1 indicates a completely correlated motion, and −1 indicates a completely anticorrelated motion.

The internal movements of CD47 exhibit certain resemblances across three complexes. First, the motions of CD47 in the complexes of the CD47 with the two variants are relatively similar. The CC’ region, encompassing residues 34–51, and the FG region, spanning residues 95–106, are found to be crucial for the binding of CD47 with SIRPαv1, SIRPαv2, and B6H12.2. Notably, these two regions exhibit highly correlated motions in all three complexes.

However, variations are also observed in the internal movements of CD47 within the three complexes. CD47 in CD47-SIRPαv1 and CD47-SIRPαv2 exhibit a mixed pattern of correlated and anticorrelated motions. The CD47 protein bound to B6H12.2 shows a significant enhancement of correlated motions compared with the CD47 proteins in CD47-SIRPαv1 and CD47-SIRPαv2. The results suggest that there is more consistency and coordination in the movement of CD47 in CD47-B6H12.2. The nature of the alterations in the dynamical correlations of CD47 is contingent upon the specific protein-binding partners that attach to it.

The CD47 proteins bound to the SIRPα variants exhibit anticorrelated motions between some different regions, while many of these regions change to correlated motions when CD47 is bound to B6H12.2. In other words, compared with the CD47 bound to SIRPαv1 and SIRPαv2, the whole protein structure of CD47 is more significantly attracted and fixed by B6H12.2. It is possible that the findings indicate a relatively strong binding affinity in the CD47-B6H12.2 pairing.

## 3. Discussion

As an important protein complex in the tumor immune response, CD47-SIRPα has great potential in the field of drug developments. In this study, we investigated the binding mechanism of the three systems, CD47-SIRPαv1, CD47-SIRPαv2, and CD47-B6H12.2. Based on the protein thermodynamic analysis and combined with the structural characteristics, a perspective can be made on the drug discovery of CD47-SIRPα. This study also provides insights into the release of the tumor immune escape caused by CD47-SIRPα.

Two avenues exist for the drug development of CD47-SIRPα, namely the creation of drugs that target CD47 and the development of drugs that target SIRPα. The results of MD simulations show that the critical residues on CD47 concentrate in the C strand and the FG fragment, which play an important role in the energy and structural analyses. The FG fragment of CD47 exhibits relatively more significant importance when it binds to SIRPαv1 or SIRPαv2. The residues Glu35, Tyr37, Glu100, Leu101, Thr102, and Arg103 in the C strand and FG fragment provide the major energy contributions of CD47 when it binds to the SIRPα variants. Among these residues, Glu35, Glu100, Leu101, and Arg103 of CD47 participate in the formation of significant hydrogen bonds or salt bridges in CD47-SIRPαv1 and CD47-SIRPαv2; the residues Tyr37, Leu101, and Arg103 of CD47 are involved in forming significant residue contacts. The drug design corresponding to CD47 could consider the inhibition of the critical residues in the C strand and the FG fragment as indicated by Site Finder.

The interaction modes of SIRPαv1 and SIRPαv2 with CD47 are quite similar. The most important feature of SIRPαv1 and SIRPαv2 in the protein binding is the grooves formed by the B2C, C’D, DE, and FG loops. The binding interfaces of the groove structures are based on the residues Leu30, Val33, Gly34, Pro35, Gln52, Lys53, Thr67, Arg69, Arg95, and Lys96 as the cores. The grooves of SIRPαv1 and SIRPαv2 are critical regions for their interactions with CD47. The C strand of CD47 forms contacts mainly with the outer edges of the grooves, and the FG fragment inserts into the groove regions. This structure contributes the major binding interfaces of CD47-SIRPαv1 and CD47-SIRPαv2. The residues Leu30, Val33, Pro35, Gln52, Lys53, Thr67, Arg69, Arg95, and Lys96 provide the major energy contributions in the SIRPα variants upon binding to CD47. Among these residues, Thr67 and Arg69 of the SIRPα variants participate in the formation of important hydrogen bonds in CD47-SIRPαv1 and CD47-SIRPαv2; Lys53, Arg69, Arg95, and Lys96 participate in the formation of stable salt bridges; and, the residues Leu30, Val33, Gln52, Lys53, Arg69, and Lys96 participate in the formation of significant residue contacts. The mechanism of the groove structures binding to CD47 could be used for drug inspiration, and small molecule drugs or cyclic peptides could be designed by imitating the critical residues. The design of inhibitors could learn the critical groove structures of the SIRPα variants to bind the C chain and FG region of CD47 so as to block the main interaction structures of CD47-SIRPαv1 and CD47-SIRPαv2. Correspondingly, according to the critical role of the C chain and FG fragment of CD47, inhibitors could be designed to bind and restrict the crucial residues of the SIRPα variants and then inhibit the protein interactions.

The antibody B6H12.2 shows higher affinity to CD47 than SIRPα variants. It is possible to obtain extraordinarily efficient CD47 inhibitors by utilizing the binding mechanism of B6H12.2 with CD47. The critical FG fragment of CD47 interacts mainly with the structural region surrounded by residues Tyr32_LC_, His92_LC_, Arg96_LC_, Thr52_HC_, Thr59_HC_, Leu100_HC_, Ala101_HC_, and Gly102_HC_ in B6H12.2. The residues Tyr32_LC_, His92_LC_, Arg96_LC_, Tyr32_HC_, Thr52_HC_, Ser53_HC_, Thr56_HC_, Thr59_HC_, Leu100_HC_, Ala101_HC_, and Gly102_HC_ provide the major energy contributions of the antibody. Among these residues, Tyr32_LC_, His92_LC_, Arg96_LC_, Tyr32_HC_, Ser53_HC_, Thr59_HC_, and Gly102_HC_ also show important influence in the structural analysis. The critical residues at these important positions could provide inspiration for the modification of antibodies and the design of macromolecular inhibitors against CD47. To explore the CD47-SIRPα inhibitors based on the interaction of CD47-B6H12.2, the critical residues are supposed to be combined, and the overall interaction mode of CD47-B6H12.2 should be considered comprehensively from the perspective of the protein-binding mechanism.

The results of our simulations could provide some guidance for the antibody optimization. The hotspot residues on the binding interface of B6H12.2 should be retained, including Tyr32_LC_, His92_LC_, Arg96_LC_, Tyr32_HC_, Thr52_HC_, Ser53_HC_, Thr56_HC_, Tyr59_HC_, Leu100_HC_, Ala101_HC_, and Gly102_HC_. Furthermore, the residues involved in significant hydrogen bonds or salt bridges should also be retained, such as Asp31_LC_, Tyr32_LC_, Lys49_LC_, Gln53_LC_, His92_LC_, Arg96_LC_, Gly31_HC_, Ser53_HC_, and Gly102_HC_. There are several residues that form close contacts with CD47 but do not exhibit a significant effect in the energetic analysis, including Phe50_LC_, Gly93_LC_, Tyr57_HC_, and Asn103_HC_. They can be mutated to form possible hydrophobic contacts, hydrogen bonds, or salt bridges and to improve the affinity of the antibody to CD47. If a residue in B6H12.2 exhibits significant hydrophilicity while its corresponding contact residue in CD47 shows obvious hydrophobicity, we can then consider converting this residue to hydrophobicity residue and vice versa. For instance, the residue Asn103_HC_ exhibits significant hydrophilicity while its corresponding contact residue Leu101 in CD47 shows obvious hydrophobicity. It may be feasible to convert the residue Asn103_HC_ in B6H12.2 to a hydrophobic residue. Furthermore, the residue Tyr57_HC_ does not show obviously charged properties while its corresponding contact residue Glu29 in CD47 possesses a negatively charged property. It may be helpful to convert the residue Tyr57_HC_ to a residue with a positive charge. These efforts may maintain or even enhance the affinity of B6H12.2 to CD47 and contribute to the development of a superior antibody.

## 4. Materials and Methods

### 4.1. Structure Preparation

The crystal structures of CD47 in complex with SIRPαv1 (CD47-SIRPαv1, entry ID: 4CMM) [16], SIRPαv2 (CD47-SIRPαv2, entry ID: 2JJS) [15], and B6H12.2 (CD47-B6H12.2, entry ID: 5TZU) [17] were retrieved from the Protein Data Bank (PDB) [36]. For these initial structures, expression tags at the terminals were deleted. Mutations and unnatural residues were mutated backed to their wild-type counterparts. The missing residues were rebuilt with MODELLER v9.25 [37,38]. The residue information at protein binding interfaces of the static complexes was summarized using PYMOL [34]. The LEaP program was used to add hydrogen atoms and counterions [39] and to immerse the proteins into truncated octahedral TIP3P water boxes, specifying a threshold value of 12.0 Å for the minimum distance between boundary solvents and proteins. The atomistic parameters were assigned with the AMBER14SB force field [39].

### 4.2. MD Simulations

To eliminate unfavorable contacts within the simulation systems, a three-staged minimization was undertaken. Firstly, we performed 1000 steps of steepest descent minimization, followed by 1500 steps of conjugate gradient minimization, with a restraint of 10.0 kcal/mol/Å^2^ applied to heavy atoms, except for water. In the second stage, we repeated the same minimization process as the first stage, but this time, we imposed a restraint of 10.0 kcal/mol/Å^2^ solely on the protein backbone atoms. Finally, 1000 steps of steepest descent minimization followed by 4000 steps of conjugate gradient minimization without any restraints were performed. The NVT ensemble was utilized to gradually heat the systems from 0 to 300 K over a duration of 30 ps. A restraint of 5.0 kcal/mol/Å^2^ was applied to the protein backbone atoms during this process. Each system was equilibrated for 300 *ps*, while the backbone atom restraints were gradually reduced from 5.0 to 0.1 kcal/mol/Å^2^. Finally, a production simulation of 500 ns was performed for each system under the NPT ensemble (*T* = 300.0 K and *P* = 1.0 atm). Temperature and pressure were regulated using the Langevin thermostat and the Monte Carlo barostat, respectively. The electrostatic interactions were computed using the particle-mesh Ewald method with a nonbonded cutoff of 9.0 Å. All bonds involving a hydrogen atom were constrained in length. The time step of the simulation was set to 2 fs. The snapshots of the simulated trajectories were recorded at intervals of 10 ps, and a total of 50,000 frames of trajectory information were obtained in a single trajectory for all complex systems. The simulations of the three systems were conducted in triplicate with the AMBER20 package [39].

### 4.3. Free Energy Calculations and Decomposition

Binding free energy (${\Delta G}_{bind}$) calculations were carried out for all the three systems of CD47-SIRPαv1, CD47-SIRPαv2, and CD47-B6H12.2. The calculations were based on the MM-GBSA method [40]. The theoretical basis can be briefly described as follows:(1)$${\Delta G}_{bind}=G_{complex}{-G}_{receptor}-G_{ligand}$$
(2)$$G=E_{gas}{\;+\;G}_{sol}-TS$$
(3)$$E_{gas}=E_{ele}+E_{vdw}$$
(4)$$G_{sol}=G_{sol,polar}+G_{sol,nonpolar}$$
(5)$$G_{sol,nonpolar}=\gamma SASA$$

Here, *G*, *T*, and *S* stand for the free energy, the temperature, and the entropy, respectively. $E_{gas}$ is the energy in vacuum, including the interactions of electrostatic ($E_{ele}$) and the van der Waals ($E_{vdw}$). The solvation free energy ($G_{sol}$) consists of the polar ($G_{sol,polar}$) and the nonpolar ($G_{sol,nonpolar}$) parts. The former was computed using the Generalized Born (GB) equation, for which the $GB_{OBC}I$ model and the mbondi2 radii were utilized. The latter was determined from a solvent-accessible surface area (SASA) dependent relation (Equation (5)) with a coefficient (*γ*) of 0.0072 kcal/mol/Å^2^. A total of 10,000 frames were extracted equidistantly from the trajectory at intervals of 5 frames, and the enthalpy change (${\Delta E}_{gas}{+\Delta G}_{sol}$) was calculated through the 10,000 snapshots of the entire trajectory for each system. Considering the large computational cost of entropy analysis, 50 snapshots were obtained from the entire trajectory at uniform intervals for each system to calculate the entropy alteration (TΔS) using the normal mode analysis. In addition, a decomposition of the binding free energy was conducted to identify energetically significant residues and hotspot regions on the proteins.

### 4.4. Trajectory Analysis

The root-mean-square deviation (RMSD) was calculated using the alpha carbon (Cα) atoms, and the root-mean-square fluctuation (RMSF) was calculated using all the heavy atoms. The protein druggable sites in the equilibrium states of simulations were analyzed using the Site Finder program in MOE [33]. The Site Finder applies the geometric method rather than energy models and takes the chemical type of receptor atoms into consideration. It is an inheritance and development from Alpha Shapes [41,42]. The methods utilized in the Site Finder include the double-linkage clustering, a connection distance of 2.5 Å, the PLB score for pocket ranking, and a refinement work provided by Volkamer in 2010 [43]. A total of 500 frames were extracted equidistantly from the trajectory to analyze the dynamical changes of proteins during the simulations. The DSSP (Definition of Secondary Structure of Protein, compiled in CPPTRAJ [44]) analysis was performed on the proteins which had undergone the structural changes in the simulations. The hydrogen bonds and salt bridges throughout the simulations were identified to characterize the important polar interactions. The hydrogen bond analysis was calculated by CPPTRAJ [44] using the geometric consideration of 3.0 Å for the distance and 135 degrees for the angle cutoff. The salt bridge analysis was carried out using the Salt Bridges Plugin in VMD1.9.3 [45].

### 4.5. Dynamical Residue Contacts Analysis

The dynamical residue contacts analysis was calculated by the native contact program in CPPTRAJ [44]. Residues located in contiguous proximity would be regarded as contacts, and these residues may have a high likelihood of forming interactions. A geometric threshold value of 5.0 Å was employed as the distance cutoff for heavy atoms. The tightness of contacts between residues was determined by analyzing their total contact fraction in the simulation trajectory. The total fraction of a specific residue pair depended on summation of atom contacts according to
(6)$$total\;fraction=\sum_{i=1}^{N}x_{i}$$
where *N* represents the total number of atom pairs forming contacts in the corresponding residue pair, *i* represents the atom pair from the two residues, and $x_{i}$ stands for the occupancy of contact time of the atom pair during the entire simulation.

### 4.6. Correlation Analysis of Dynamical Motions

The dynamical cross-correlation matrix (DCCM) analyses were applied to the three CD47 proteins in the three molecular complexes, CD47-SIRPαv1, CD47-SIRPαv2, and CD47-B6H12.2 to estimate the relative coordinate fluctuations of correlation and anti-correlation during simulations. The DCCM analyses were calculated through protein Cα atoms, and the motions of Cα atoms could represent the motion correlations of corresponding protein residues. The correlation or anti-correlation here can be explained as the coordinate or incoordinate motions in CD47 during simulations. The covariance was calculated according to
(7)$$c(x,y)=<(x\;-\;\overset{-}{x})\;\cdot \;(y\;-\;\overset{-}{y})>$$
where *c* represents the covariance of two residues. The *x* and *y* represent the coordinates of two different residues, respectively. The numerical value of DCCM was calculated according to
(8)$$C(x,y)=\frac{c(x,y)}{\sqrt{c(x,x)\cdot c(y,y)}}$$
where *C* indicates that multiplying the covariance matrix by the correlation coefficient to obtain the dynamic cross-correlation. The above calculations were performed for all the residues to obtain the matrices of DCCM.

## 5. Conclusions

In this study, we systematically revealed the binding mechanism of CD47-SIRPαv1, CD47-SIRPαv2, and CD47-B6H12.2 by MD simulations. The three complexes exhibited favorable stability and convergence during the 500 ns simulations. The characteristic that CD47-B6H12.2 possess a higher affinity than CD47-SIRPαv1 and CD47-SIRPαv2 was consistently verified in the free energy calculations. Subsequent residue energy decomposition and analyses of the hydrogen bonds, salt bridges, and dynamical residue contacts significantly revealed the energetic and structural characteristics of the hotspot residues in proteins. According to our simulations, the crucial residues of CD47 situate mainly in the C strand and FG region when it interacts with the SIRPα variants, including Glu35, Tyr37, Leu101, Thr102, and Arg103. The DCCM analysis reveals that the CD47 protein exhibits more consistent motions when it binds to B6H12.2, indicating that the initial structure of CD47 could be attracted and immobilized to a greater extent by the antibody. The residues Leu30, Val33, Gln52, Lys53, Thr67, Arg69, Arg95, and Lys96 of SIRPαv1 and SIRPαv2 possess important contributions on the protein interaction, which distribute in the crucial groove structures formed by the B2C, C’D, DE, and FG loops. There are obvious druggable sites in the groove regions, and the C’D loops on the binding interfaces have undergone notable dynamical changes throughout the simulations. The critical residues of B6H12.2 contain Tyr32_LC_, His92_LC_, Arg96_LC_, Tyr32_HC_, Thr52_HC_, Ser53_HC_, Ala101_HC_, and Gly102_HC_. Based on the binding mechanisms of CD47-SIRPαv1, CD47-SIRPαv2, and CD47-B6H12.2, the meaningful understandings and inspirations on the drug discovery targeting CD47-SIRPα could be obtained. Simultaneously, the information could provide valuable insights for releasing the tumor immune escape caused by CD47-SIRPα.

## Figures and Tables

**Figure 1 molecules-28-04610-f001:**
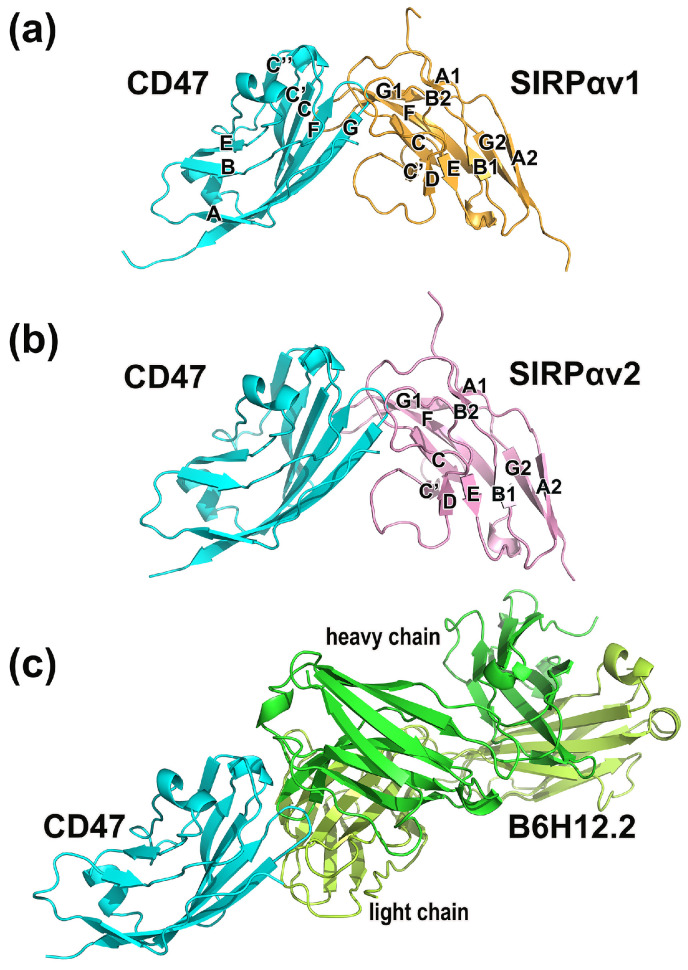
The three molecular complexes simulated in this study. (**a**) CD47-SIRPαv1, (**b**) CD47-SIRPαv2, and (**c**) CD47-B6H12.2. The β-sheets in the variants are labeled with capital letters.

**Figure 2 molecules-28-04610-f002:**
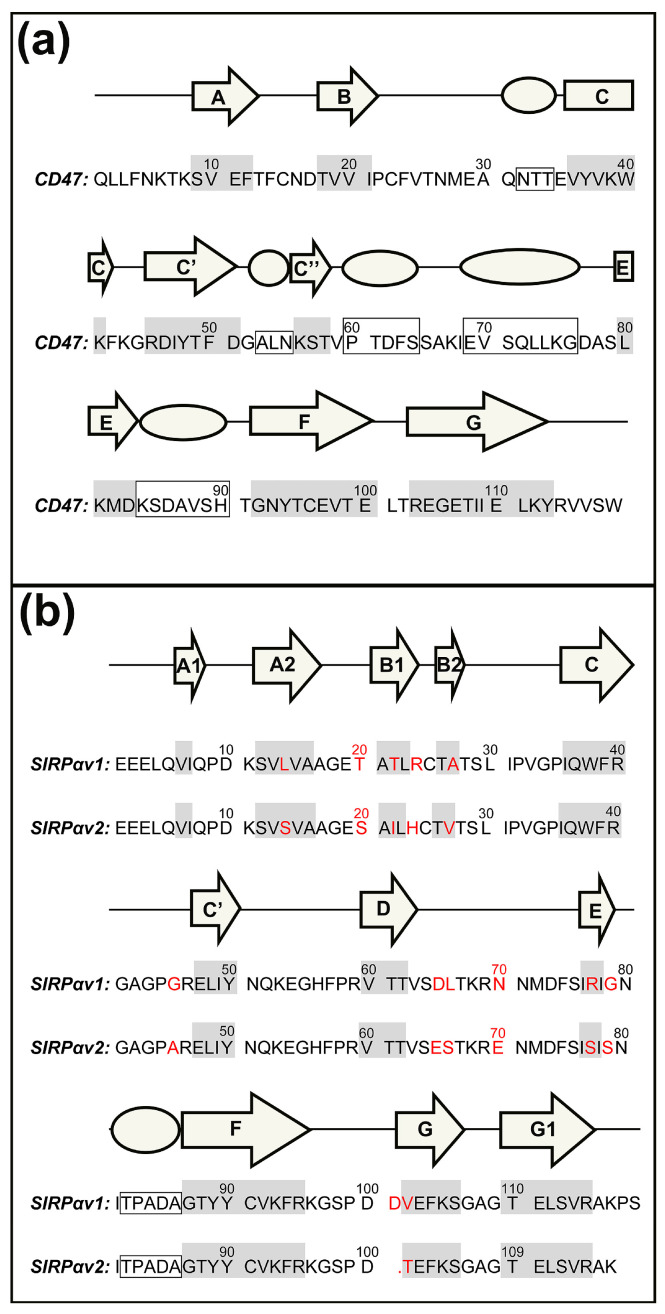
The sequences and the secondary structures of (**a**) CD47 and (**b**) SIRPαv1 and SIRPαv2. The β-sheets are marked with shadings on the sequences and with the arrows above, and α-helices are marked with the squares and the spheres. The 13 mutant residues between SIRPαv1 and SIRPαv2 are highlighted.

**Figure 3 molecules-28-04610-f003:**
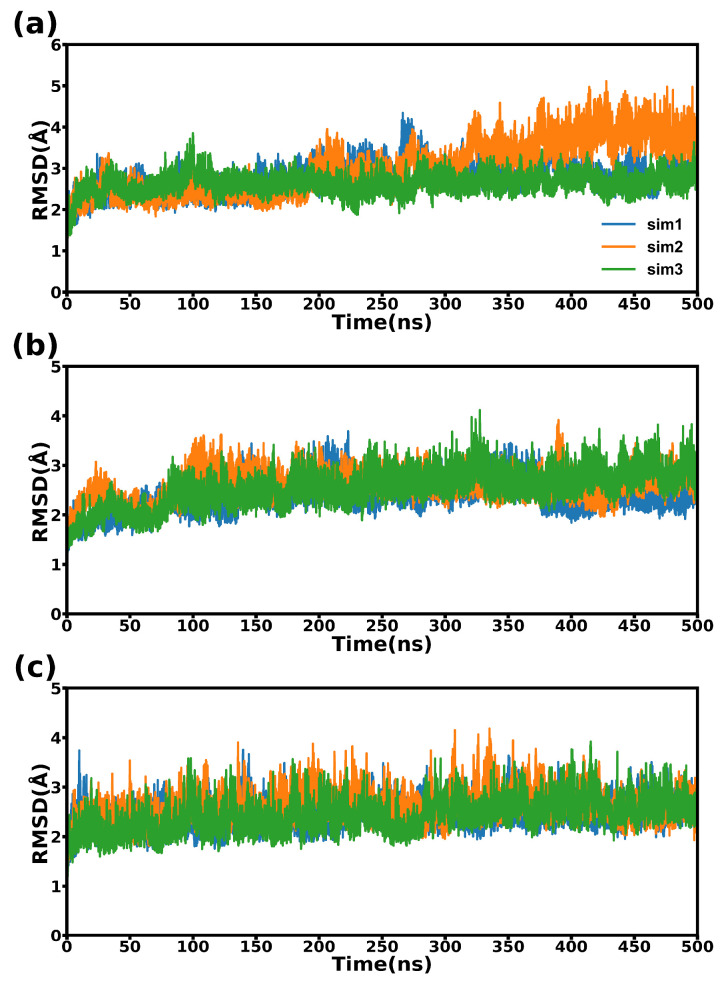
The RMSD curves of the three complexes. (**a**) CD47-SIRPαv1, (**b**) CD47-SIRPαv2, and (**c**) CD47-B6H12.2. The values on the abscissas represent the simulation time. The repetitions of the simulations are indicated with different lines.

**Figure 4 molecules-28-04610-f004:**
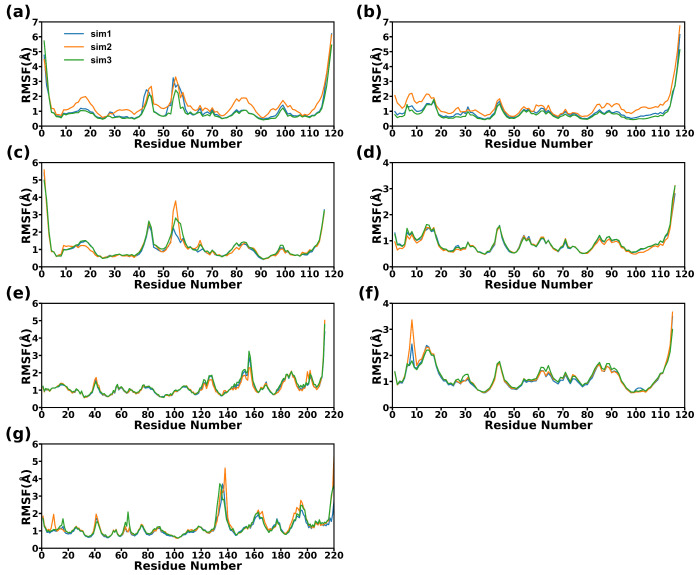
The RMSF curves of the complexes. (**a**) SIRPαv1, (**b**) CD47 bound to SIRPαv1, (**c**) SIRPαv2, (**d**) CD47 bound to SIRPαv2, (**e**) light chain of B6H12.2, (**f**) CD47 bound to B6H12.2, and (**g**) heavy chain of B6H12.2. The values on the abscissas represent the residue numbers in the proteins. The repetitions of the simulations are indicated with different lines.

**Figure 5 molecules-28-04610-f005:**
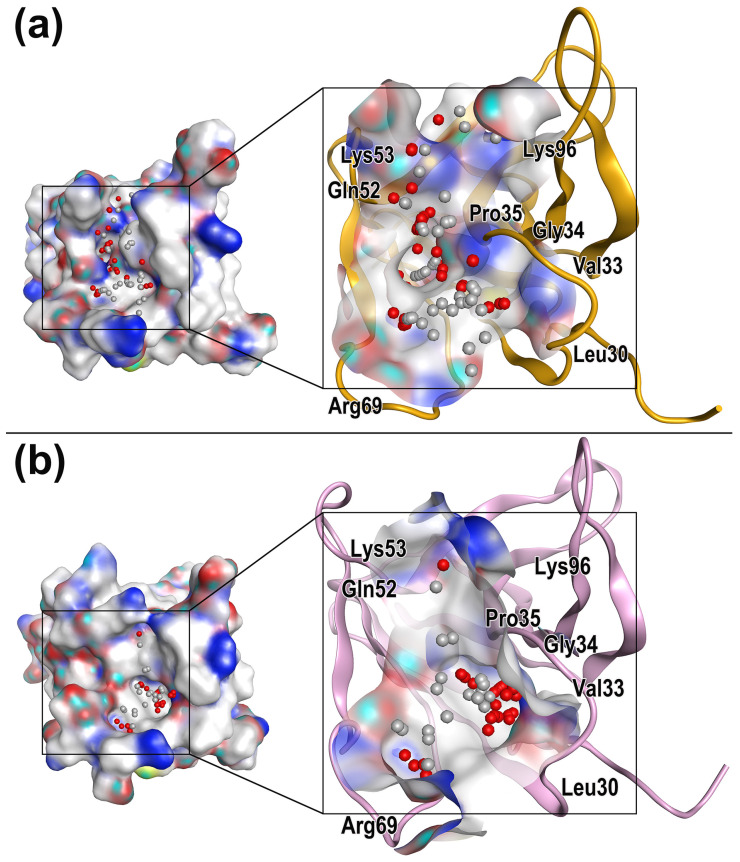
Characterization of druggable site in (**a**) SIRPαv1 and (**b**) SIRPαv2. The druggable sites and surrounding residues concentrate mainly in the groove regions formed by the B2C, C’D, DE, and FG loops of SIRPαv1 and SIRPαv2.

**Figure 6 molecules-28-04610-f006:**
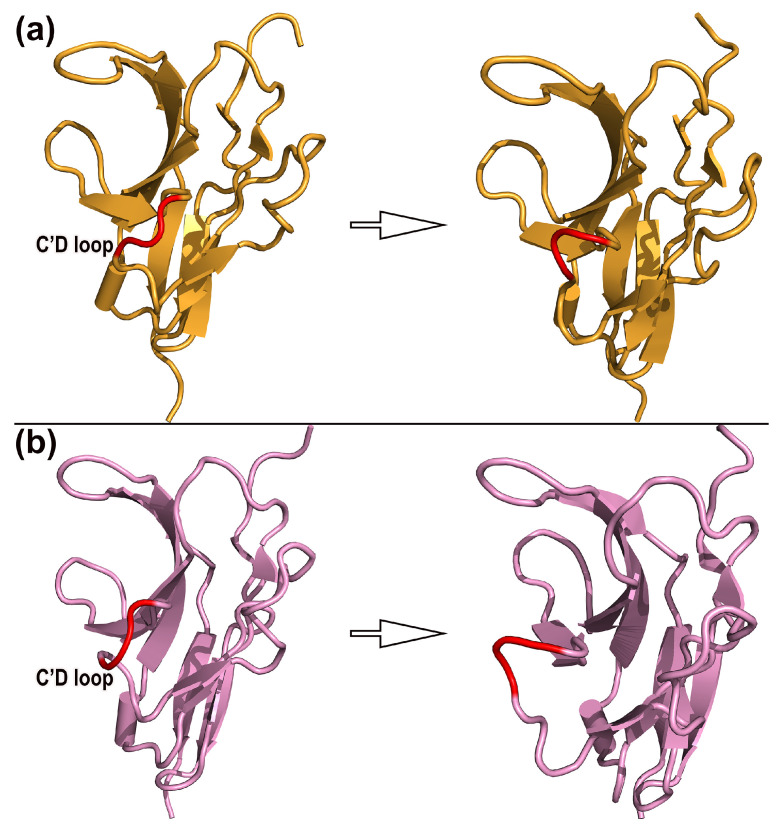
Dynamical changes of the C’D loops throughout the simulations in (**a**) SIRPαv1 and (**b**) SIRPαv2. The left half of the figure shows the structural state at the beginning of the simulation, and the right half shows the equilibrium structural state at the end. The C’D loops are important components of the critical groove region on the binding interfaces of SIRPαv1 and SIRPαv2. The map shows results of the first simulation, while results of the second and third simulations are shown in Appendix A.

**Figure 7 molecules-28-04610-f007:**
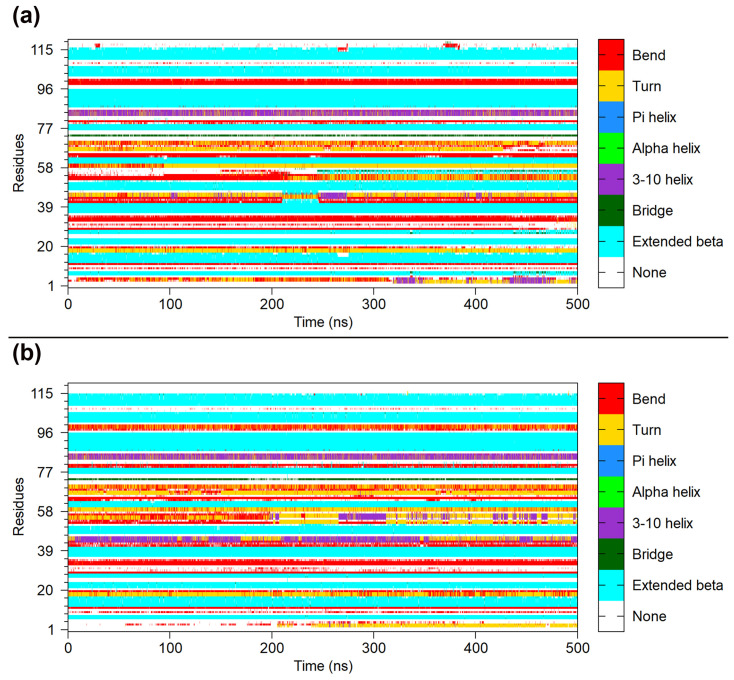
The DSSP analysis of (**a**) SIRPαv1 and (**b**) SIRPαv2. This figure shows the DSSP analysis for SIRPαv1 and SIRPαv1 in the first simulation, while results of the second and third simulations are shown in the Appendix A.

**Figure 8 molecules-28-04610-f008:**
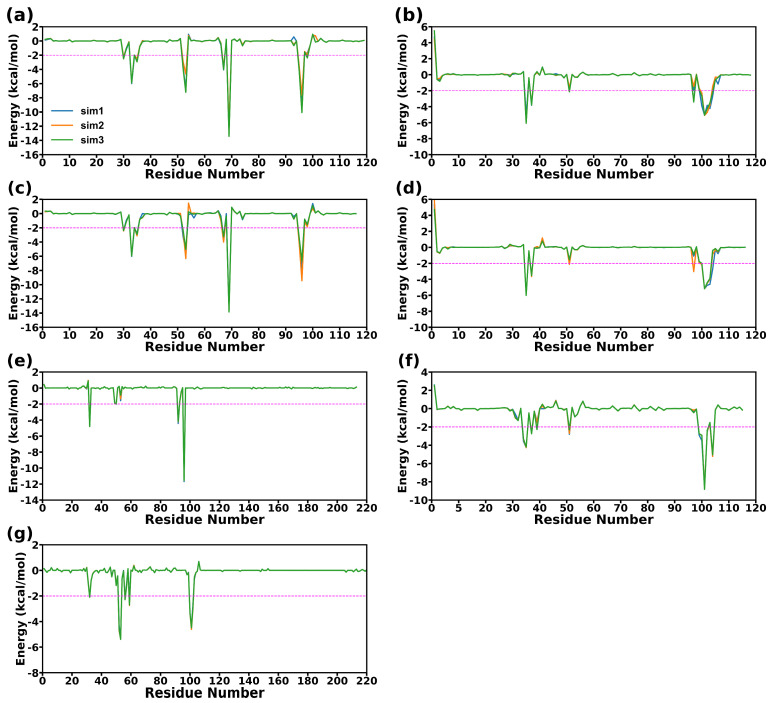
The residue free energy decomposition curves of different proteins. (**a**) SIRPαv1, (**b**) CD47 bound to SIRPαv1, (**c**) SIRPαv2, (**d**) CD47 bound to SIRPαv2, (**e**) light chain of B6H12.2, (**f**) CD47 bound to B6H12.2, and (**g**) heavy chain of B6H12.2. The values on the abscissas represent the residue numbers. The repetitions of the simulations are indicated with different lines.

**Figure 9 molecules-28-04610-f009:**
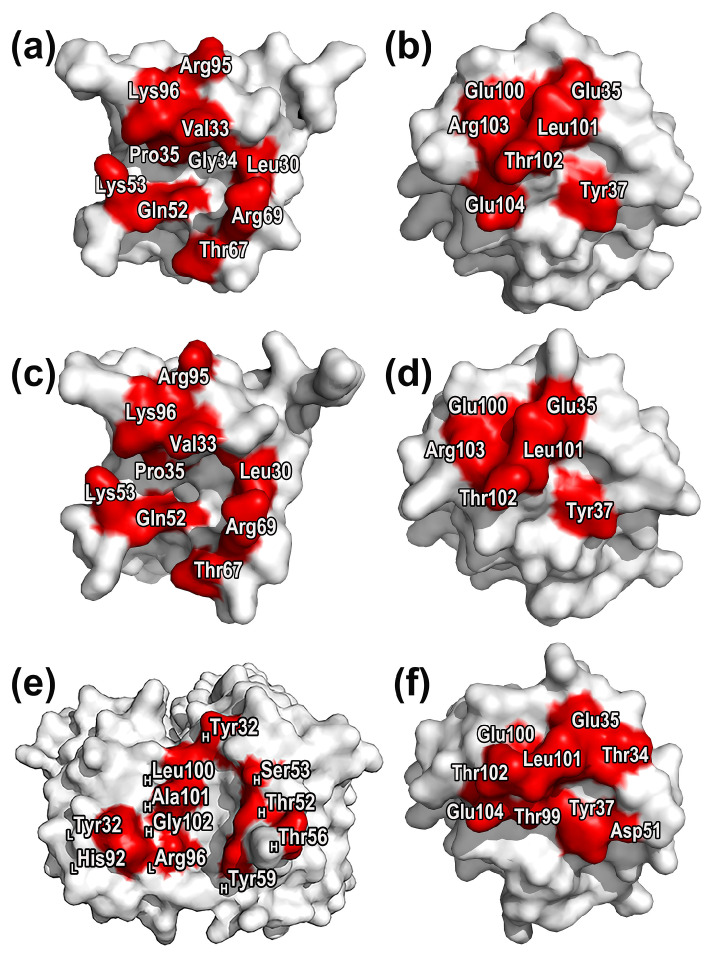
The hotspot residues of the proteins in the complexes. (**a**) SIRPαv1, (**b**) CD47 bound to SIRPαv1, (**c**) SIRPαv2, (**d**) CD47 bound to SIRPαv2, (**e**) B6H12.2, and (**f**) CD47 bound to B6H12.2. The free energy decomposition criterions for hotspot residues are lower than −2 kcal/mol in at least two replicates. The hotspot residues are labeled with their residue names and numbers.

**Figure 10 molecules-28-04610-f010:**
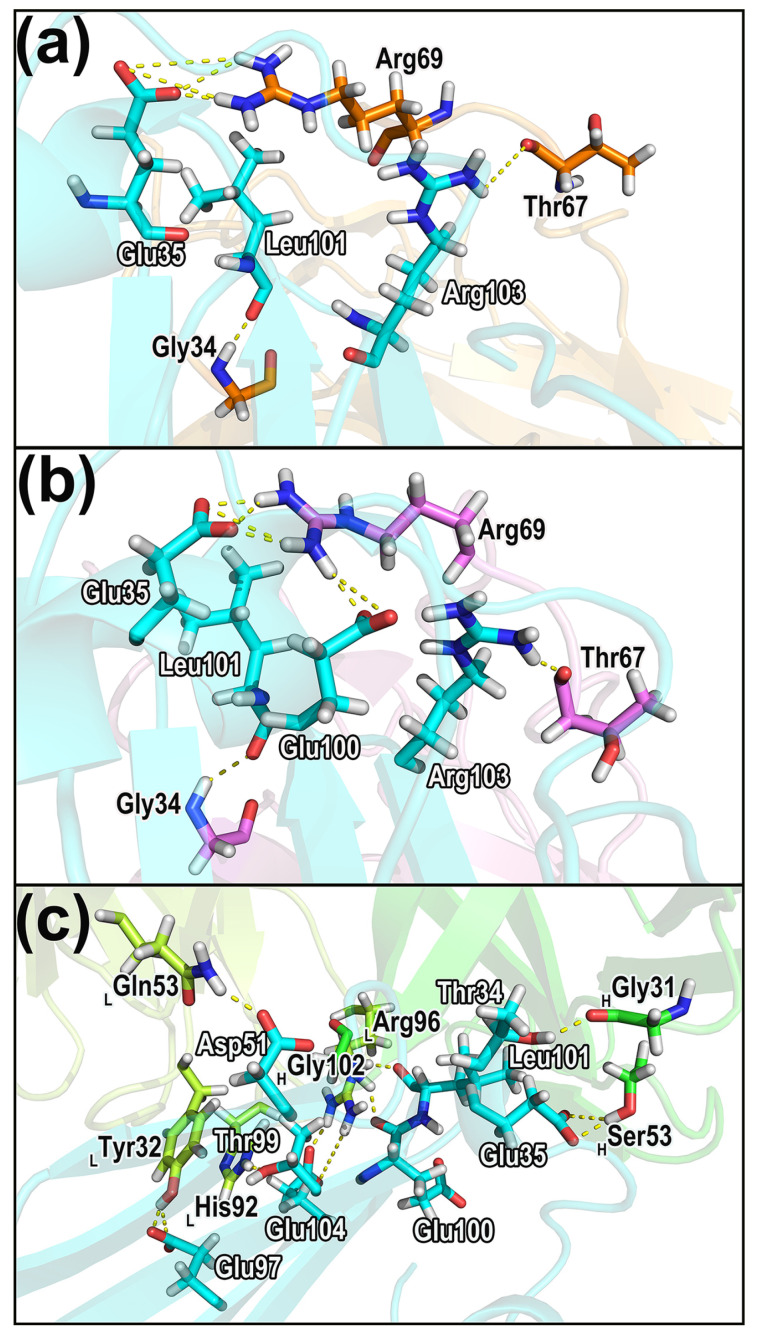
The hydrogen bonds that contribute significantly to the binding of the complexes. (**a**) CD47-SIRPαv1. (**b**) CD47-SIRPαv2. (**c**) CD47-B6H12.2. The hydrogen bonds labeled in the figure appear in all of the triplicates, and their time percentages are greater than 40% in at least two replicates. The corresponding residues in CD47 are labeled in white, and the residues of its binding partners are labeled in black. The subscript “L” and “H” represent the light and the heavy chain, respectively.

**Figure 11 molecules-28-04610-f011:**
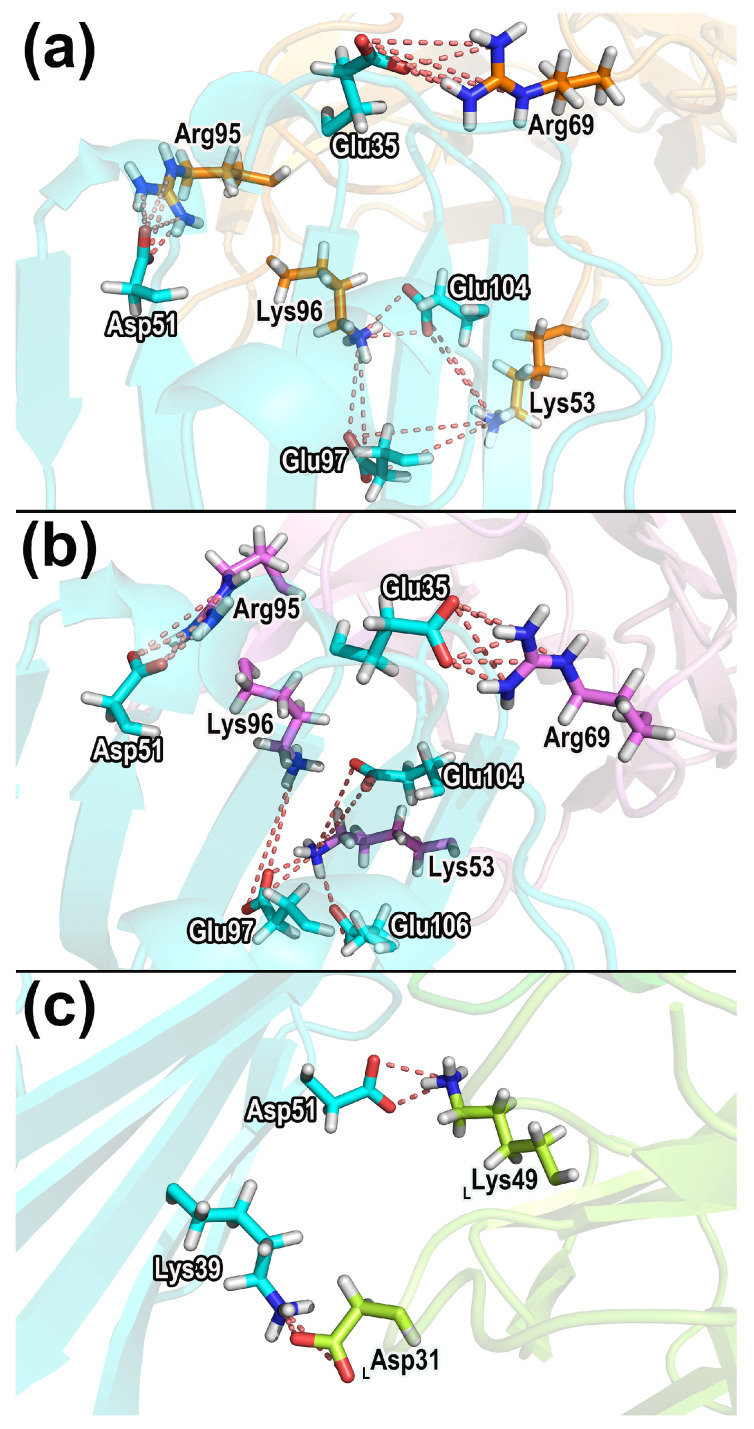
The significant salt bridges in (**a**) CD47-SIRPαv1, (**b**) CD47-SIRPαv2, and (**c**) CD47-B6H12.2. The salt bridges labeled in the figure appear in all salt bridge analyses of the triplicate, and their time percentages are greater than 20% in at least two replicates.

**Figure 12 molecules-28-04610-f012:**
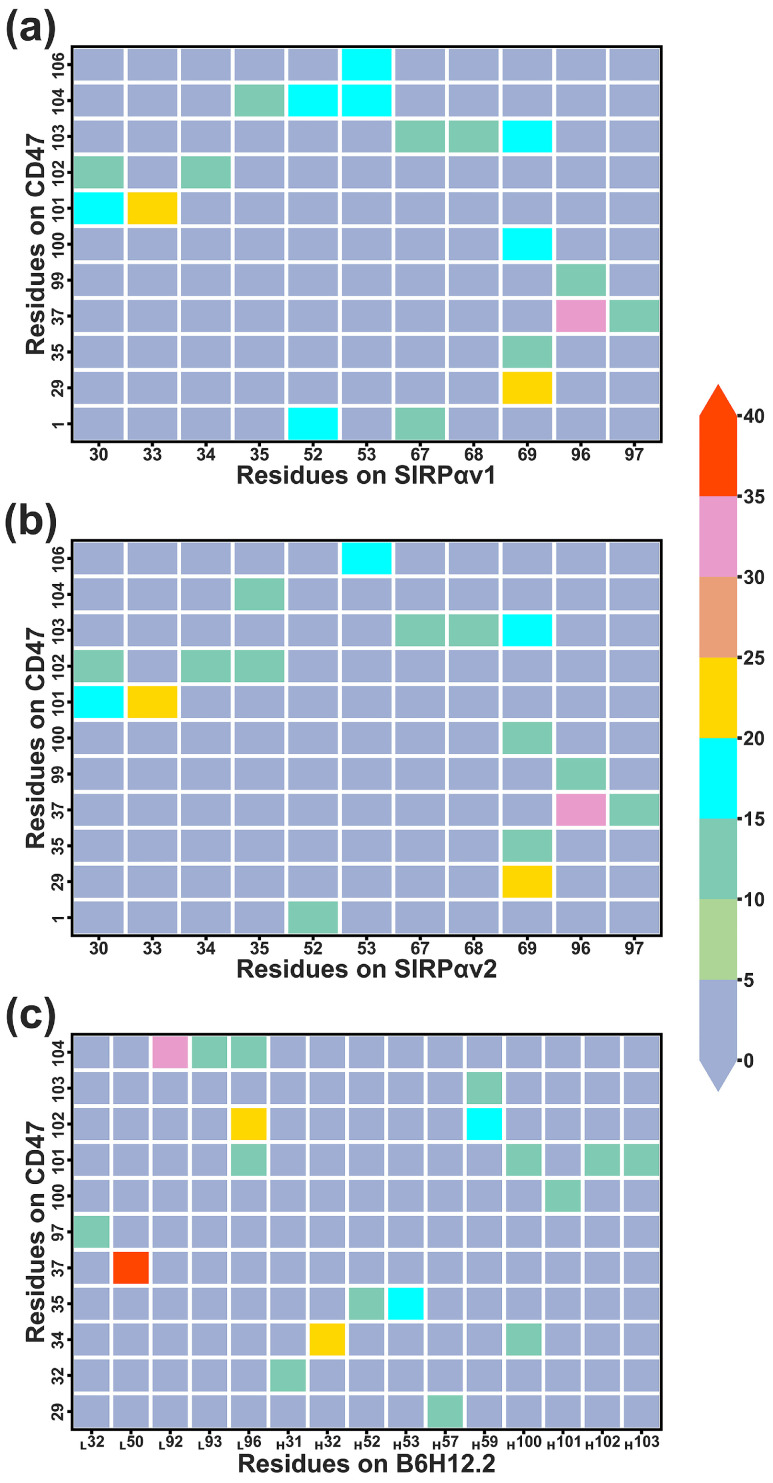
The dynamical residue–residue contact maps for (**a**) CD47-SIRPαv1, (**b**) CD47-SIRPαv2, and (**c**) CD47-B6H12.2. The numerical values of the color bar represent fractions obtained by the native contact calculations. The values indicate tightness and stability of the dynamical contacts between residue pairs, and the higher values represent the tighter contacts. The map shows results of the first simulation, while results of the second and third simulations are shown in the Appendix A.

**Figure 13 molecules-28-04610-f013:**
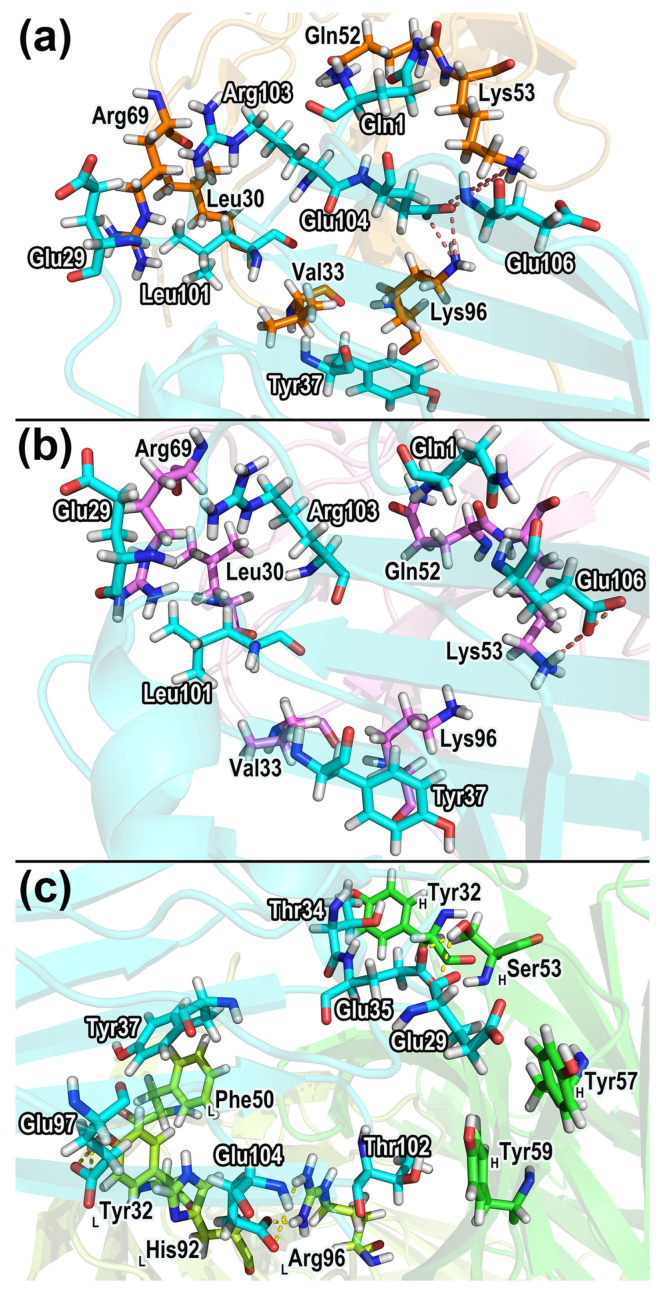
The prominent residue contacts in the dynamical interactions of proteins. (**a**) CD47-SIRPαv1, (**b**) CD47-SIRPαv2, and (**c**) CD47-B6H12.2. The dynamical residue contacts labeled in the figure appear in all contact calculations of the triplicate, and their total fractions are greater than 14 in at least two replicates. The important hydrogen bonds (colored in yellow) and electrostatic interactions (colored in red) are indicated with dash lines. The threshold value of 14 is chosen as the cut off according to overall situations of the residue contacts.

**Figure 14 molecules-28-04610-f014:**
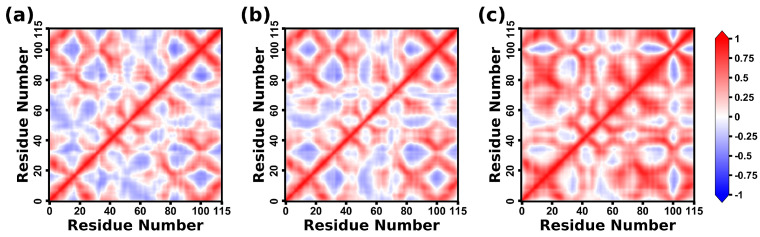
The DCCM maps for the CD47 proteins in (**a**) CD47-SIRPαv1, (**b**) CD47-SIRPαv2, and (**c**) CD47-B6H12.2. The motion correlation is represented by a value between −1 and +1, and the higher values indicate the stronger correlations. The maps on both sides of each DCCM diagonal are symmetrical. This map shows results of the first simulation, while results of the second and third simulations are shown in the Appendix A.

**Table 1 molecules-28-04610-t001:** Contact residues in the static structures of CD47/SIRPαv1, CD47/SIRPαv2, and CD47/B6H12.2.

CD47 Residues	Partner Residues ^a^
**CD47-SIRPαv1:**	
Gln1, Leu2, Leu3, Lys6, Asn27, Glu29, Ala30, Gln31, Thr34, Glu35, Val36, Tyr37, Lys39, Asp46, Thr49, Glu97, Thr99, Glu100, Leu101, Thr102, Arg103, Glu104, Gly105, Glu106	Ser29, Leu30, Ile31, Pro32, Val33, Gly34, Pro35, Ile36, Gln37, Tyr50, Asn51, Gln52, Lys53, Glu54, Leu66, Thr67, Lys68, Arg69, Phe74, Lys93, Lys96, Gly97, Ser98, Pro99, Asp100, Asp101
**CD47-SIRPαv2:**	
Gln1, Leu2, Leu3, Lys6, Asn27, Glu29, Ala30,Gln31, Thr34, Glu35, Val36, Tyr37, Lys39, Lys41, Asp46, Thr49, Glu97, Thr99, Glu100, Leu101, Thr102, Arg103, Glu104, Gly105, Glu106	Ser29, Leu30, Ile31, Pro32, Val33, Gly34, Pro35, Ile36, Gln37, Tyr50, Asn51, Gln52, Lys53, Glu54, His56, Ser66, Thr67, Lys68, Arg69, Glu70, Phe74, Lys93, Lys96, Gly97, Ser98, Pro99, Asp100
**CD47-B6H12.2 ^b^:**	
Met28, Glu29, Ala30, Gln31, Asn32, Thr34, Glu35, Val36, Tyr37, Lys39, Asp46, Asp51, Ala53, Leu54, Glu97, Thr99, Glu100, Leu101, Thr102, Arg103, Glu104	Ser30_LC_, Asp31_LC_, Tyr32_LC_, Lys49_LC_, Phe50_LC_, Gln52_LC_, Gly91_LC_, His92_LC_, Gly93_LC_, Phe94_LC_, Arg96_LC_, Ser30_HC_, Gly31_HC_, Tyr32_HC_, Gly33_HC_, Trp47_HC_, Thr50_HC_, Ile51_HC_, Thr52_HC_, Ser53_HC_, Gly54_HC_, Gly55_HC_, Thr56_HC_, Tyr57_HC_, Tyr59_HC_, Ser99_HC_, Leu100_HC_, Ala101_HC_, Gly102_HC_, Asn103_HC_

^a^ The binding partners of CD47 are SIRPαv1, SIRPαv2, and B6H12.2. ^b^ The “LC” means the light chain in B6H12.2 and the “HC” means the heavy chain.

**Table 2 molecules-28-04610-t002:** The Binding Free Energies of CD47 to the Binding Partners Calculated by MM-GBSA ^a^.

Contribution	CD47-SIRPαv1	CD47-SIRPαv2	CD47-B6H12.2
Δ*E_ele_*	−557.87 ± 72.76	−522.69 ± 69.19	−549.61 ± 29.91
Δ*E_vdw_*	−89.88 ± 6.72	−86.09 ± 8.99	−97.56 ± 6.01
Δ*E_gas_* ^b^	−647.74 ± 73.77	−608.78 ± 72.85	−647.17 ± 29.57
Δ*G_sol-polar_*	575.80 ± 68.36	540.53 ± 63.17	565.76 ± 26.28
Δ*G_sol-np_*	−14.54 ± 0.79	−13.76 ± 1.35	−13.99 ± 0.58
Δ*G_sol_* ^c^	561.27 ± 67.84	526.78 ± 62.13	551.77 ± 26.09
Δ*H_tot_* ^d^	−86.48 ± 9.72	−82.00 ± 13.26	−95.40 ± 7.85
*T*Δ*S* ^e^	−53.75 ± 6.78	−54.55 ± 5.78	−51.76 ± 7.24
Δ*G_bind_* ^f^	−32.73	−27.45	−43.64
*K_d_*	0.74 ± 0.07 μM	0.64 ± 0.06 μM	0.5 nM

^a^ The values after the “±” signs indicate the standard deviations. This table shows results of the first simulation, while results of the second and third simulations are shown in the Appendix A. The unit of Binding Free Energies is “kcal/mol”. ^b^ Δ*Egas* = *Einternal* + *Eele* + *Evdw*. ^c^ Δ*Gsol* = *Esol-np* + *Esol-polar*. ^d^ Δ*Htot* = Δ*Egas* + Δ*Gsol*, the enthalpy change. ^e^ *T*Δ*S*, entropy change. ^f^ Δ*Gbind* = Δ*Htot* − *T*Δ*S*, the Gibbs free energy.

**Table 3 molecules-28-04610-t003:** The hydrogen bonds at the protein binding interfaces of CD47-SIRPαv1, CD47-SIRPαv2, and CD47-B6H12.2 ^a^.

Acceptor Residues	Acceptor Atoms	Donor Residues	Donor Atoms	Fraction 1 ^b^	Fraction 2	Fraction 3
**CD47-SIRPαv1:**						
SIRPα_Thr67	O	CD47_Arg103	NH1	88.39%	84.93%	91.16%
CD47_Leu101	O	SIRPα_Gly34	N	79.75%	73.11%	80.38%
CD47_Glu35	OE2	SIRPα_Arg69	NH2	49.75%	49.16%	52.81%
CD47_Glu35	OE1	SIRPα_Arg69	NH2	49.22%	50.90%	48.36%
CD47_Glu35	OE1	SIRPα_Arg69	NH1	34.85%	45.86%	43.36%
CD47_Glu35	OE2	SIRPα_Arg69	NH1	27.67%	41.40%	47.19%
**CD47-SIRPαv2:**						
SIRPα_Thr67	O	CD47_Arg103	NH1	86.75%	90.19%	85.80%
CD47_Leu101	O	SIRPα_Gly34	N	85.18%	83.24%	83.27%
CD47_Glu35	OE1	SIRPα_Arg69	NH2	50.61%	48.62%	52.99%
CD47_Glu35	OE2	SIRPα_Arg69	NH2	51.46%	52.00%	47.72%
CD47_Glu35	OE2	SIRPα_Arg69	NH1	51.63%	46.76%	49.83%
CD47_Glu100	OE1	SIRPα_Arg69	NH1	49.31%	45.71%	41.79%
CD47_Glu100	OE2	SIRPα_Arg69	NH1	37.97%	41.74%	45.84%
CD47_Glu35	OE1	SIRPα_Arg69	NH1	39.36%	44.43%	41.80%
**CD47-B6H12.2:**						
CD47_Thr99	OG1	LC_His92	NE2	77.36%	71.15%	67.44%
CD47_Glu100	O	HC_Gly102	N	76.87%	80.10%	80.90%
HC_Gly31	O	CD47_Thr34	OG1	70.27%	63.52%	68.58%
CD47_Asp51	OD2	LC_Gln53	NE2	62.09%	53.64%	36.49%
CD47_Glu104	OE1	LC_Arg96	NH1	55.32%	53.87%	41.68%
CD47_Leu101	O	LC_Arg96	NE	54.75%	55.88%	49.23%
CD47_Glu35	OE1	HC_Ser53	OG	52.38%	53.57%	34.55%
CD47_Glu35	OE2	HC_Ser53	OG	46.38%	45.29%	63.99%
CD47_Glu97	OE2	LC_Tyr32	OH	50.82%	41.96%	48.33%
CD47_Glu97	OE1	LC_Tyr32	OH	42.93%	50.97%	46.96%
CD47_Glu104	OE2	LC_Arg96	NH1	41.36%	44.13%	55.27%

^a^ The important hydrogen bonds forming in three complexes, CD47-SIRPαv1, CD47-SIRPαv2, and CD47-B6H12.2. The hydrogen bond displayed in each entry of the table appears in all of the repetitions, and the fractions are greater than 40% in at least two replicates. LC: light chain, HC: heavy chain. ^b^ The “Fraction” means the fraction data in the n-th simulation. The fraction in each entry is a percentage value representing the time occupancy of corresponding hydrogen bond over the entire simulation.

**Table 4 molecules-28-04610-t004:** Analysis of the salt bridges in CD47/SIRPαv1, CD47/SIRPαv2, and CD47/B6H12.2 ^a^.

Negative Residues ^b^	Positive Residues	Fraction 1 ^c^	Fraction 2	Fraction 3
**CD47-SIRPαv1:**				
CD47_Glu35	SIRPαv1_Arg69	32.66%	43.16%	45.53%
CD47_Glu97	SIRPαv1_Lys96	18.49%	32.07%	74.37%
CD47_Glu97	SIRPαv1_Lys53	45.35%	15.85%	40.15%
CD47_Asp51	SIRPαv1_Arg95	36.37%	35.15%	42.20%
CD47_Glu104	SIRPαv1_Lys96	78.59%	31.45%	4.53%
CD47_Glu104	SIRPαv1_Lys53	4.59%	24.19%	48.76%
**CD47-SIRPαv2:**				
CD47_Glu35	SIRPαv2_Arg69	50.01%	49.77%	52.45%
CD47_Asp51	SIRPαv2_Arg95	41.00%	44.15%	35.85%
CD47_Glu97	SIRPαv2_Lys96	17.95%	59.08%	36.13%
CD47_Glu104	SIRPαv2_Lys53	18.07%	34.29%	35.60%
CD47_Glu106	SIRPαv2_Lys53	27.74%	20.61%	38.96%
CD47_Glu97	SIRPαv2_Lys53	24.54%	43.30%	5.95%
**CD47-B6H12.2:**				
CD47_Asp51	LC_Lys49	99.56%	99.77%	99.78%
LC_Asp31	CD47_Lys39	28.65%	35.78%	60.68%

^a^ The salt bridges in CD47-SIRPαv1, CD47-SIRPαv2, and CD47-B6H12.2. The salt bridges displayed in each entry appear in all three simulations, and its total fractions are greater than 20% in at least two simulations. LC: light chain, HC: heavy chain. ^b^ The oxygen atoms in the side chains of negative (acidic) residues could form salt bridges with the nitrogen atoms in the side chains of positive (basic) residues. ^c^ The “Fraction” means the fraction data in the n-th simulation. The fractions in the residue pairs represent the time percentages of a specific definition, which means that the distances between the center of oxygen atoms in acidic residue and the center of nitrogen atoms in basic residue were within 3.5 Å.

## Data Availability

The simulation parameters of CD47-SIRPαv1, CD47-SIRPαv2 and CD47-B6H12.2 are available at https://github.com/Ks1997cris/CD47-SIRP (accessed on 3 June 2023), including the initial files of the MD simulations and the corresponding parameter code settings.

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
