# Peer review of "Binding Mechanism of CD47 with SIRPα Variants and Its Antibody: Elucidated by Molecular Dynamics Simulations"

_molecules, 2023, doi:10.3390/molecules28124610_

Round 1
Reviewer 1 Report
Huang et al. have performed molecular dynamics simulations to reveal the interactions taking place between CD47 and two SIRPa variants, as well as the anti-CD47 monoclonal antibody B6H12.2. Through this they aim to characterize the binding mechanism and to identify hot spot residues involved in the interactions. The CD47-SIRPa interaction has gathered much interest, especially from the drug design point of view, for which structural studies are essential. The authors of this paper have done an extensive and thorough job with the molecular dynamics simulations and subsequent structural analyses. The statements and conclusions drawn by the authors are coherent and supported by the listed citations. The work is also within the scope of the journal.
- You have identified druggable sites in the equilibrium states of simulations using MOE. It could be good to provide more information on how they were detected (at least in the methods section). Do they have a score that helped you to choose, or you just identified this by looking at them? Could you find them also with other software for druggable site prediction or are there perhaps publications you can cite, which would show the identification of these sites?
- In 2.3 you also say that “The druggable sites and surrounding residues of SIRPαv1 and SIRPαv2 are similar, which indicates that there may be no significant difference in the drug properties of the two proteins”. When you check Figure 5, especially the smaller, overall surface picture of the two SIRPa, they seem to have a different surface shape, groove size and potentially also charge distribution. This is something that could affect drug specificity, so I would have liked a bit more thorough explanations around the figure and discussion related to the impact on drug design.
- In figure 11, it would be good to show the electrostatic interactions taking place between the different residues, rather than just showing the residues. The same comment goes for figure 13, where I would like to see the hydrogen bonds and the electrostatic interactions.
Overall, the manuscript could benefit from a final language check to get rid of small mistakes here and there. I have tried to catch some of them here, but I might not have gotten them all.
- Change all “was provided/was shown in figure” to “is provided/is shown”
- In the abstract “The C’D loops on the binding interfaces undergone”, change undergone to undergo
- Next sentence after the “The C’D loops on the binding interfaces” has “structural impactions”, change impactions to impact
- “The N-terminal domain of SIRPα exhibits highly polymorphism”, change highly to high
- “CD47 is a critical role for almost all solid tumor cells”, change “is a critical role” to “has a critical role” or “is critical”
Author Response
Response to Reviewer #1:
Comment: Huang et al. have performed molecular dynamics simulations to reveal the interactions taking place between CD47 and two SIRPa variants, as well as the anti-CD47 monoclonal antibody B6H12.2. Through this they aim to characterize the binding mechanism and to identify hot spot residues involved in the interactions. The CD47-SIRPa interaction has gathered much interest, especially from the drug design point of view, for which structural studies are essential. The authors of this paper have done an extensive and thorough job with the molecular dynamics simulations and subsequent structural analyses. The statements and conclusions drawn by the authors are coherent and supported by the listed citations. The work is also within the scope of the journal.
Response: We are grateful for the reviewer's recognition of our efforts and their comments on our manuscript.
- You have identified druggable sites in the equilibrium states of simulations using MOE. It could be good to provide more information on how they were detected (at least in the method section). Do they have a score that helped you to choose, or you just identified this by looking at them? Could you find them also with other software for druggable site prediction or are there perhaps publications you can cite, which would show the identification of these sites?
Response: Thank you very much for pointing our negligence in the interpretation of the druggable site analysis.
The protein druggable sites in the equilibrium states of simulations were analyzed using the Site Finder program in MOE [1]. The Site Finder applies the geometric method rather than energy models, and the chemical type of receptor atoms are taken into consideration. It is an inheritance and development from Alpha Shapes [2, 3]. The methods utilized in the Site Finder include the double-linkage clustering, a connection distance of 2.5 Å, the PLB score for pocket ranking and a refinement work provided by Volkamer in 2010 [4].
- St. West, S., 2022.02 Molecular Operating Environment (MOE). Chemical Computing Group ULC, 1010 Sherbooke 2023,910.
- Edelsbrunner, H. In Alpha Shapes — a Survey, 2009.
- Edelsbrunner, H.; Facello, M.; Fu, P.; Liang, J., Measuring proteins and voids in proteins. In Proceedings of the 28th Hawaii International Conference on System Sciences, IEEE Computer Society: 1995; p 256.
- Volkamer, A.; Griewel, A.; Grombacher, T.; Rarey, M., Analyzing the topology of active sites: on the predictionof pockets and subpockets. J Chem Inf Model 2010,50, (11), 2041-52.
These sentences and references were added to the revised manuscript.
- In 2.3 you also say that “The druggable sites and surrounding residues of SIRPαv1 and SIRPαv2 are similar, which indicates that there may be no significant difference in the drug properties of the two proteins”. When you check Figure 5, especially the smaller, overall surface picture of the two SIRPα, they seem to have a different surface shape, groove size and potentially also charge distribution. This is something that could affect drug specificity, so I would have liked a bit more thorough explanations around the figure and discussion related to the impact on drug design.
Response: We are sorry for not presenting the analysis of the druggable site well in the Figure 5.
The difference of the druggable sites of SIRPαv1 and SIRPαv2 as shown in the Figure 5 may be attributed to the randomization of the structural conformations, which were extracted from the last frame of the simulation. In fact, the properties of the druggable sites of the two proteins in the simulated steady state are very similar. The similarity of the drug properties is based on the consistencies of the overall structures, the critical residues in the druggable sites, the energetic and the structural features in SIRPαv1 and SIRPαv2. We believe that in the 500 ns simulation, by considering the stable conformation as a whole structure, the drug properties of the two variants will be convergence.
The inappropriate rendering angles and inconsistent placement may mislead the readers. Therefore, we regenerated the figure with the same orientation and annotation mode.
- In figure 11, it would be good to show the electrostatic interactions taking place between the different residues, rather than just showing the residues. The same comment goes for figure 13, where I would like to the hydrogen bonds and the electrostatic interactions.
Response: Thank you so much for point out this issue. We updated these two figures. The hydrogen bonds and electrostatic interactions
- Change all “was provided/was shown in figure” to “is provided/is shown”.
Response: Thank you for the suggestion. We checked the manuscript and corrected the expressions, including other grammatical or expressive errors.
- In the abstract “The C’D loops on the binding interfaces undergone”, change undergone to undergo.
Response: Thanks for the suggestion. The expression was corrected.
- Next sentence after the “The C’D loops on the binding interfaces” has “structural impactions”, change impactions to impact.
Response: Thank you. We corrected this error.
- “The N-terminal domain of SIRPα exhibits highly polymorphism”, change “highly” to “high”.
Response: Thanks! We changed “highly polymorphism” to “high polymorphism”.
- “CD47 is a critical role for almost all solid tumors cells”, change “is a critical role” to “has a critical role” or “is critical”.
Response: Thanks for the suggestion. We have changed all “is a critical role” to “has a critical role”.
Author Response
Response to Reviewer #2:
Comment: Authors should explain how their findings could contribute the development of a superior antibody compared to the well-known B6H12.2 antibody.
Response: Thank you so much for the suggestion.
To develop a superior antibody, the hotspot residues on the binding interface of B6H12.2 should be reserved, including Tyr32L, His92L, Arg96L, Tyr32H, Thr52H, Ser53H, Thr56H, Tyr59H, Leu100H, Ala101H, and Gly102H; Besides, the residues involving in significant hydrogen bonds or salt bridges should also be retained, which include Asp31L, Tyr32L, Lys49L, Gln53L, His92L, Arg96L, Gly31H, Ser53H, and Gly102H. On the other hand, there are several residues that form close contacts with CD47 but not exhibit significant effect in the energetic analysis, including Phe50L, Gly93L, Tyr57H, and Asn103H. They can be mutated to form possible hydrophobic contacts, hydrogen bonds or salt bridges and to improve the affinity of the antibody to CD47. For instance, if a residue in B6H12.2 exhibits significant hydrophilicity while its corresponding contact residue in CD47 shows obvious hydrophobicity, then we could consider to convert this residue to hydrophobicity residue, and vice versa.
These sentences were added into the revised manuscript.
Are there any alternative regions on the natural partner that could potentially elicit a stronger immune response or improve the antibody’s neutralization capabilities?
Response: It is a good idea to target the CD47’s natural partner, the SIRPα. However, it is difficult to de novo design a high-efficient antibody that requires several rounds of experimental and theoretical study.
I would suggest authors discuss potential engineering strategies that could be employed to optimize the B6H12.2 antibody based on the insights gained from their MD simulations. For instance, they can explore the feasibility of introducing specific mutations or modifications in the antibody informed by their findings in MD simulations and explore its affinity with the CD47.
Response: Thank you for the suggestion. As we stated in the response to the first issue, we identified the residues Phe50L, Gly93L, Tyr57H, and Asn103H of B6H12.2 that form close contact with CD47 while they do not show significant effect in the energetic analysis, hydrogen bonds or salt bridges. Among them, the residues Asn103H exhibits significant hydrophilicity while its corresponding contact residue Leu101 in CD47 shows obvious hydrophobicity. It may be effective to convert the residue Asn103H in B6H12.2 to hydrophobicity residue. Tyr57H does not show obviously charged property while its corresponding contact residue Glu29 in CD47 possesses negatively charged property. It may be helpful to convert the residue Tyr57H to the residue with positive charges.
The potential engineering strategies that could be employed to optimize the B6H12.2 antibody, based on the insights gained from their MD simulations, are provided in the discussion section of the revised manuscript.